# Incentivizing Data Collection from Heterogeneous Clients in Federated Learning

## Abstract

Federated learning (FL) provides a promising paradigm for facilitating collaboration between multiple clients that jointly learn a global model without directly sharing their local data. However, existing research suffers from two caveats: 1) From the perspective of agents, voluntary and unselfish participation is often assumed. But self-interested agents may opt out of the system or provide low-quality contributions without proper incentives; 2) From the mechanism designer's perspective, the aggregated models can be unsatisfactory as the existing game-theoretical federated learning approach for data collection ignores the potential heterogeneous effort caused by contributed data.

To alleviate above challenges, we propose an incentive-aware framework for agent participation that considers data heterogeneity to accelerate the convergence process. Specifically, we first introduce the notion of Wasserstein distance to explicitly illustrate the heterogeneous effort and reformulate the existing upper bound of convergence. To induce truthful reporting from agents, we analyze and measure the generalization error gap of any two agents by leveraging the peer prediction mechanism to develop score functions. We further present a two-stage Stackelberg game model that formalizes the process and examines the existence of equilibrium. Extensive experiments on real-world datasets demonstrate the effectiveness of our proposed incentive mechanism.

## 1 Introduction

Real-world dataset is often distributed across multiple locations and its amalgamation into a centralized repository for training is often hindered by privacy considerations. In response to the constraints presented by data limitations across various locations, federated learning (FL) provides a promising paradigm for facilitating collaboration between multiple agents (also called clients or devices) that jointly learn a global model without directly sharing their local data. In the standard FL framework (McMahan et al., 2017), each individual agent performs model training using its local data, and a central server (called the learner) aggregates the local models from agents into a single global model.

Although federated learning is a rapidly growing research area with observed success in improving speed, efficiency, and accuracy (Li et al., 2019; Yang et al., 2021), many studies have made an implicit and unrealistic assumption that agents will voluntarily spend their resources on collecting their local data and training their local models to help the central server to refine the global model. In practice, without proper incentives, self-interested agents may opt out of contributing their resources to the system or provide low-quality models (Liu & Wei, 2020; Blum et al., 2021; Wei et al., 2021; Karimireddy et al., 2022; Wei et al., 2022). The literature has observed recent efforts on providing incentives for FL agents to contribute model training based on sufficient amounts of local data (Donahue & Kleinberg, 2021b;a; Hasan, 2021; Cui et al., 2022; Cho et al., 2022). However, we argue that a **common limitation** of these studies is that they measure the contribution of each agent by the sample size used for training the uploaded model and incentivize agents to use more data samples. However, relying solely on the sample size as a measure of contribution may not accurately capture the quality of data samples. It is known that aggregating local models trained from a large amount of data does not always achieve fast convergence of highly accurate global models if those data are biased (McMahan et al., 2017; Zhao et al., 2018). In practice, local data on different agents are heterogeneous – they are not independently and identically distributed (non-iid). Hence, the appropriate method to incentivize agents should take into account not only the size of the sample but also

the potential data heterogeneity. **Such a solution is necessary for designing incentive mechanisms in a practical FL system but currently missing from all existing work** (Blum et al., 2021; Liu & Wei, 2020; Karimireddy et al., 2022; Zhou et al., 2021; Donahue & Kleinberg, 2021a;b; Yu et al., 2020; Cho et al., 2022). In this work, we seek to answer the following pertinent question:

> *How do we incentivize agents (local clients) to provide local models that are trained on high-quality data and achieve fast convergence on a highly accurate global model?*

To answer this, we propose an incentive-aware FL framework that considers data heterogeneity to accelerate the convergence process. Our key contributions are summarized as follows.

- We propose to use Wasserstein distance to quantify the degree of non-iid in the local training data of an agent. We are the first to prove that Wasserstein distance is an important factor in the convergence bound of FL with heterogeneous data (Lemma 4.2). [Section 4]
- We present a two-stage Stackelberg game model for the incentive-aware framework, and prove the existence of equilibrium (Theorem 5.2). [Section 5]
- To induce truthful reporting, we design scoring (reward or payment) functions to reward agents using a measure that we developed to quantify the generalization error gap of any two agents (Theorem 4.3). [Section 5]
- Experiments on real-world datasets demonstrate the effectiveness of the proposed mechanism in terms of its ability to incentivize improvement. We further illustrate the existing equilibrium that no one can unilaterally reduce his invested effort level to obtain a higher utility. [Section 6]

## 2  RELATED WORK

The incentive issue has been listed as an outstanding problem in federated learning (McMahan et al., 2017). Recent works have touched on the challenge of incentive design in federated learning (Pang et al., 2022; Zhou et al., 2021; Yu et al., 2020), while they overlook the agents' strategic behaviors induced by insufficient incentives. We summarize existing research on this game-theoretic research.

**Coalition game.** Recent efforts (Donahue & Kleinberg, 2021a;b; Cho et al., 2022; Hasan, 2021; Cui et al., 2022) have sought to conceptualize federated learning through the lens of coalition games involving self-interested agents, i.e., exploring how agents can optimally satisfy their individual incentives defined differently from our goal. Rather than focusing on training a single global model, they consider the scenario where each client seeks to identify the most advantageous coalition of agents to federate, with the primary aim of minimizing its own error.

**Contribution evaluation.** Another line of relevant work (Han et al., 2022; Kang et al., 2019; Zhang et al., 2021; Khan et al., 2020) aims to incentivize agents to contribute resources for FL and promote long-term participation by providing monetary compensation for their contributions, determined using game-theoretic tools. Certain prior studies have attempted to quantify the contributions of agents using metrics like accuracy or loss per round. However, these works fail to analyze model performance theoretically, with some neglecting the effects of data heterogeneity on models (Kong et al., 2022) and others overlooking the impact of samples on data distributions (Xu et al., 2021; Wang et al., 2020). Our work provides a theoretical link between data heterogeneity and models, offering insights into data heterogeneity and helping us handle data diversity more effectively.

**Data collection.** Having distinct objectives, (Cho et al., 2023a) and (Cho et al., 2023b) strive to directly maximize the number of satisfied agents utilizing a universal model. In contrast, Blum et al. (Blum et al., 2021) and Karimireddy et al. (Karimireddy et al., 2022) explore an orthogonal setting, where each client seeks to satisfy its constraint of minimal expected error while concurrently limiting the number of samples contributed to FL. While these works establish essential insights for mean estimation and linear regression problems, their applicability to complex ML models (i.e., neural networks) is limited. Also, these works also neglect the discrepancy of data distributions among different agents. We address these limitations by evaluating the non-iid degree of the data contributed by agents and creating score functions with peer prediction to quantify agents' contributions.

## 3  PROBLEM FORMULATION

We first introduce the preliminaries of the FL model and formulate the game-theoretical problem.

### 3.1 PRELIMINARIES

**The distributed optimization model.** Consider a distributed optimization model with $F_k : \mathbb{R}^d \to \mathbb{R}$, such that $\min_{\mathbf{w}} \quad F(\mathbf{w}) \triangleq \sum_{k=1}^{N} p_k F_k(\mathbf{w})$, where $\mathbf{w}$ is the model parameters to be optimized, $N$ denotes the number of agents, $p_k$ indicates the weight of the $k$-th agent such that $p_k \geq 0$ and $\sum_{k=1}^{N} p_k = 1$. Usually, $p_k$ is set to be the proportion of the number of data points contributed by agent $k$, *i.e.*, $p_k = |\mathcal{D}_k| / \sum_{i=1}^{N} |\mathcal{D}_k|$, where $\mathcal{D}_k$ is agent $k$'s local dataset. Each agent participates in training with the local objective $F_k(\boldsymbol{w})$, defined as: $F_k(\mathbf{w}) \triangleq \mathbb{E}_{\zeta^k \sim \mathcal{D}_k}[l(\mathbf{w}, \zeta^k)]$, where $l(\cdot, \cdot)$ represents the loss function, and datapoint $\zeta^k = (\boldsymbol{x}, y)$ uniformly sampled from dataset $\mathcal{D}_k$.

**Assumptions.** Here, we present some generalized basic assumptions in convergence analysis. Note that the first two assumptions are standard in convex/non-convex optimization (Li et al., 2019; Yang et al., 2022; Wang et al., 2019; Koloskova et al., 2020). The assumptions of bounded gradient and local variance are both standard (Yang et al., 2021; 2022; Stich, 2018).

**Assumption 3.1.** *($\mu$-Strongly Convex). There exists a constant $\mu > 0$, for any $\mathbf{v}, \mathbf{w} \in \mathbb{R}^d$, $F_k(\mathbf{v}) \geq F_k(\mathbf{w}) + \langle \nabla F_k(\mathbf{w}), \mathbf{v} - \mathbf{w} \rangle + \frac{\mu}{2} \|\mathbf{v} - \mathbf{w}\|_2^2, \forall k \in [N]$.*

**Assumption 3.2.** *(L-Lipschitz Continuous). There exists a constant $L > 0$, for any $\mathbf{v}, \mathbf{w} \in \mathbb{R}^d$, $F_k(\mathbf{v}) \leq F_k(\mathbf{w}) + \langle \nabla F_k(\mathbf{w}), \mathbf{v} - \mathbf{w} \rangle + \frac{L}{2} \|\mathbf{v} - \mathbf{w}\|_2^2, \forall k \in [N]$.*

**Assumption 3.3.** *(Bounded Local Variance). The variance of stochastic gradients for each agent is bounded: $\mathbb{E}[\|\nabla F_k(\mathbf{w}; \zeta^k) - \nabla F_k(\mathbf{w})\|^2] \leq \sigma_k^2, \forall k \in [N]$.*

**Assumption 3.4.** *(Bounded Gradient on Random Sample). The stochastic gradients on any sample are uniformly bounded,* i.e., *$\mathbb{E}[\|\nabla F_k(\mathbf{w}_t^k; \zeta^k)\|^2] \leq G^2, \forall k \in [N]$, and epoch $t \in [T-1]$.*

**Quantifying the non-iid degree (heterogeneous effort).** Empirical evidence (McMahan et al., 2017; Zhao et al., 2018) revealed that the intrinsic statistical challenge of FL, *i.e.*, data heterogeneity, will result in reduced accuracy and slow convergence of the global model in an FL system. However, many prior works (Wang et al., 2019; Koloskova et al., 2020) make an assumption to bound the gradient dissimilarity with constants, which did not explicitly illustrate this heterogeneous effort. We propose quantifying the degree of non-iid (data heterogeneity) using the notion of *Wasserstein distance*. Specifically, the Wasserstein distance is defined by the probability distance for the discrete data distribution on agent $k$ compared with the potential distribution for all agents,

$$\delta_k = \frac{1}{2} \sum_{i=1}^{I} |p^{(k)}(y = i) - p^{(c)}(y = i)|,$$

where $I$ represents the number of classes in a classification problem, $p^{(c)}$ indicates the reference (iid) data distribution in the centralized setting, and $p^{(k)}$ denotes the local distribution of agent $k$. Note that $\delta_k \in [0, 1]$ describes the "cost" of transforming agent $k$'s data distribution into the actual reference data distribution. We let $\boldsymbol{\delta} = \{\delta_1, \delta_2, ..., \delta_N\}$ be the set of non-iid degrees of all agents.

### 3.2 GAME-THEORETIC FORMULATION

We start with a standard FL scenario which consists of the learner and a set of agents $[N]$, each with a dataset $\mathcal{D}_k$ sampled from its corresponding heterogeneous data distribution $p^{(k)}$. From the learner's perspective, it is natural for her to aim for faster model convergence by minimizing the heterogeneity across data distributions. To gain a better understanding, we assume that the learner is willing to design well-tailored payment functions that can evaluate the contribution of each agent and provide appropriate rewards. Agents strive to attain maximum utility in this scenario.

We describe the whole scenario as a game with the following two entities (players):

- *Learner*: seeking to train a classifier that endeavors to encourage agents to decrease the degree of non-iid. Thereby, attaining faster convergence at a reduced cost. To accomplish this, the learner incentivizes agents through the use of payment functions as a reward for their efforts.
- *Agent*: investing its efforts $e_k$ in gathering data with the ultimate aim of maximizing the utility (payoff). It is reasonable to assume that each agent incurs a cost while investing its efforts, denoted by the cost function $cost_k(e_k) : [0, 1] \to \mathbb{R}^+$.

To establish a formal representation, we denote the endeavor of an agent to achieve a more homogeneous local distribution as $e_k \in [0,1], \forall k$. Assume that $\delta_k(e_k) : [0,1] \to [0,1]$ is a non-increasing function with the effort coefficient $e_k$. For example, $e_k = 0$ indicates that the agent $k$ will keep current data; $e_k = 1$ depicts the scenario where the agent $k$ makes his own effort to meet the requirement of data homogeneity.

In the above formulated game, we want to further make an assumption that it is a complete information game, wherein all agents possess comprehensive knowledge about each other, as well as the learner's established payment functions, and her objectives. Additionally, the learner is privy to complete information about the agents, including their cost functions. Following this, we present the definition of utility/payoff functions of both players and the corresponding optimization problems.

**Definition 3.1.** *(Agent's Utility). For agent $k$, its utility can be denoted by $u_k(e_k) \triangleq \text{Payment}_k(e_k) - \text{Cost}_k(e_k), \forall k \in [N]$, where $\text{Payment}_k(e_k)$ is the payment function designed by the learner, and $\text{Cost}_k(e_k)$ is agent $k$'s cost function. Intuitively, the payment and cost functions increase with effort coefficient $e_k$.*

**Definition 3.2.** *(The learner's Payoff). For a learner, the payoff of developing an FL model is $\text{Payoff} \triangleq \text{Reward}(e) - \sum_{k=1}^{N} \text{Payment}_k(e_k)$, where $\text{Reward}(e)$ represents the corresponding reward given the global model's performance under a set of agents' effort levels $e = \{e_1, e_2, \cdots, e_N\}$. Intuitively, the faster the global model converges, the more reward.*

**Problem 1** (Maximization of agent's utility). For any agent $k$, its goal is to maximize utility through $\max_{e_k \in [0,1]} \quad u_k(e_k) = \text{Payment}_k(e_k) - \text{Cost}_k(e_k)$.

**Problem 2** (Maximization of learner's payoff). Correspondingly, the learner's goal is to maximize its payoff conditioned on Incentive Compatibility (IC), *i.e.*, satisfying the utility of agents are non-negative. The whole optimization problem can be shown as follows.

$$\max_{e} \quad \text{Payoff}, \qquad \text{s.t.} \quad u_k(e_k) \geq 0, e_k \in [0,1], \forall k, k' \in [N].$$

Although the definition of utility or payoff functions is conceptually straightforward, the tough task is to establish connections between agents and the learner by leveraging the non-iid metric. In the proceeding section, we shall delve into the theoretical aspects to uncover potential connections.

## 4 PREPARATION: CONVERGENCE ANALYSIS OF FEDERATED LEARNING

In this section, we reformulate the existing upper bound of convergence using the non-iid metric $\delta_k$ (Example 1), which mathematically connects agents and the learner, implicitly allowing one to manipulate its non-iid degree to speed up the global model's convergence. Besides, we analyze and measure the generalization error gap between any two agents by leveraging the peer prediction mechanism to induce truthful reporting.

### 4.1 RETHINKING CONVERGENCE BOUND USING WASSERSTEIN DISTANCE

Even though a long line of recent research focuses on the convergence bound of federated learning within different settings (for example, partial/full agent participation with a specific non-iid metric (Li et al., 2019), bounded gradient/Hessian dissimilarity (Karimireddy et al., 2020), B-local dissimilarity (Li et al., 2020), uniformly bounded variance (Yu et al., 2019), asynchronous communications & different local steps per agent (Yang et al., 2022)), the generic form is similar. The fundamental idea is to leverage the following Lemma 4.1 to derive the convergence results.

**Lemma 4.1.** *(Connection with Lemma 3.1 in (Stich, 2018)). Let $\left\{\mathbf{w}_t^k\right\}_{t \geq 0}$ be defined as the parallel sequences of rounds (epochs) obtained by performing SGD, and $\overline{\mathbf{w}}_t = \frac{1}{N} \sum_{k=1}^{N} \mathbf{w}_t^k$. Let $\overline{\mathbf{g}}_t = \sum_{k=1}^{N} p_k \nabla F_k(\mathbf{w}_t^k)$ represent the weighted sum of gradients of loss functions $F_k$ for client $k$. Then, $\mathbf{g}_t = \sum_{k=1}^{N} p_k \nabla F_k(\mathbf{w}_t^k, \xi_t^k)$. Let $F(\cdot)$ be $L$-smooth and $\mu$-strongly convex and the learning rate $\eta_t \leq \frac{1}{4L}$. Then*

$$\mathbb{E} \|\overline{\mathbf{w}}_{t+1} - \mathbf{w}^\star\|^2 \leq (1 - \mu\eta_t) \mathbb{E} \|\overline{\mathbf{w}}_t - \mathbf{w}^\star\|^2 + \eta_t^2 \mathbb{E} \|\mathbf{g}_t - \overline{\mathbf{g}}_t\|^2 - \frac{1}{2}\eta_t \mathbb{E} \left(F\left(\overline{\mathbf{w}}_t\right) - F(\mathbf{w}^*)\right)$$

$$+ 2\eta_t \mathbb{E}[\sum_{k=1}^{N} p_k \|\overline{\mathbf{w}}_t - \mathbf{w}_t^k\|^2]. \tag{1}$$

Here, we focus on the general *divergence term* $\mathbb{E}[\sum_{k=1}^{N} p_k ||\overline{\mathbf{w}}_t - \mathbf{w}_t^k||^2]$ (W.l.o.g, $p_k$ denotes the weight of $k$-th agent) throughout the convergence analysis due to the observation that the discrepancy of models incurred by data heterogeneity will be shown up as the divergence of model parameters (Zhao et al., 2018). Therefore, the divergence term implicitly encodes the impact of data heterogeneity. Thus, we are trying to build the connection between convergence and the Wasserstein distance by analyzing the divergence between aggregated model through FedAvg and any agent $k$'s model ($\mathbf{w}_t^k$). Here, we present the main result of our paper, given in Lemma 4.2.

**Lemma 4.2.** *(Bounded the divergence of $\{\mathbf{w}_t^k\}$). Let Assumption 3.4 hold and G be defined therein, given the synchronization interval $E$ (local epochs), the learning rate $\eta_t$. Suppose that $\nabla_{\mathbf{w}} \mathbb{E}_{\boldsymbol{x}|y=i}[\ell_i(\boldsymbol{x}, \mathbf{w})]$ is $L_{\boldsymbol{x}|y=i}$-Lipschitz, it follows that*

$$\mathbb{E}[\sum_{k=1}^{N} p_k ||\overline{\mathbf{w}}_t - \mathbf{w}_t^k||^2] \leq 16(E-1)G^2\eta_t^2(1+2\eta_t)^{2(E-1)} \sum_{k=1}^{N} p_k \overbrace{\left( \sum_{i=1}^{I} |p^{(k)}(y=i) - p^{(c)}(y=i)| \right)^2}^{4\delta_k^2}.$$

The existing convergence bound could be rewritten by substituting the upper bound of the model divergence term $\mathbb{E}[\sum_{k=1}^{N} p_k ||\overline{\mathbf{w}}_t - \mathbf{w}_t^k||^2]$ with its new form. Here, we introduce an illustrative example (Example 1) that highlights convergence results, which helps develop our utility/payoff function in Section 5.

**Example 1.** *(Connection with Theorem 1 in (Li et al., 2019)). Given Assumptions 3.1 to 3.3, suppose $\sigma_k^2$, $G$, $\mu$, $L$ are defined therein, the synchronization interval $E$ (local epochs), $\kappa = \frac{L}{\mu}$, $\gamma = \max\{8\kappa, E\}$ and the learning rate $\eta_t = \frac{2}{\mu(\gamma+t)}$. FedAvg with full device participation follows*

$$\mathbb{E}[F(\mathbf{w}_T)] - F^* \leq \frac{\kappa}{\gamma + T - 1}(\frac{2B}{\mu} + \frac{\mu\gamma}{2}\mathbb{E}||\mathbf{w}_1 - \mathbf{w}^*||^2).$$

*where $B = \sum_{k=1}^{N} p_k^2 \sigma_k^2 + 6L\Gamma + \underbrace{16(E-1)G^2\eta_t^2(1+2\eta_t)^{2(E-1)} \sum_{k=1}^{N} p_k \delta_k^2}_{\text{Upper bound shown in Lemma 4.2}}.$*

In Example 1, the upper bound of convergence (aka, model performance) is affected by the non-iid metric $\delta_k^2$, which means one builds a mathematical link between agents and the learner, which implicitly allows one to manipulate its non-iid degree to speed up the global model's convergence. The result highlighted in Example 1 gives us the certainty that the non-iid metric can be effectively utilized to develop the payoff function.

In this paper, we study the convergence results in a generic form about the term $\mathbb{E}[\sum_{k=1}^{N} p_k ||\overline{\mathbf{w}}_t - \mathbf{w}_t^k||^2]$. Therefore, the exploration of the other terms in the convergence results is not in our scope. For interested readers, please refer to (Li et al., 2019) for more details. Leveraging this generic form, our results maintain broad applicability without compromising the precision of prior findings. This is evidenced in Lemma 4.2, with additional examples provided in Appendix A.6. Besides, to further show broad applicability, in Appendix A.7, we further discuss how to use $\delta_k$ to rewrite another common assumption (Yang et al., 2021; 2022; Wang et al., 2019; Stich, 2018), which bounds gradient dissimilarity or variance using constants.

## 4.2 GENERALIZATION ERROR GAP BETWEEN PEERS

Note that in federated learning, the learner only receives the responses provided by agents (model $\mathbf{w}$) without any further information. Our goal is to leverage these responses to construct utility functions that promote truthful reporting. We propose to leverage the peer prediction mechanism (Liu & Wei, 2020; Shnayder et al., 2016), for evaluating responses and ensuring appropriate compensation.

Suppose that the learner owns an auxiliary/test dataset $\mathcal{D}_c = \{(x_n, y_n)\}_{n=1}^{N}$ following a reference data distribution $p^{(c)}$. To start with, we consider *one-shot* setting that agent $k$, $k'$ submit their models only in a single communication round $E$.

**Theorem 4.3.** *Suppose that the expected loss function $F_c(\cdot)$ also follows Assumption 3.2, then the upper bound of the generalization error gap between agent $k$ and $k'$ is*

$$F_c(\mathbf{w}^k) - F_c(\mathbf{w}^{k'}) \leq \Phi\delta_k^2 + \Phi\delta_{k'}^2 + \Upsilon \tag{2}$$

*where $\Phi = 16L^2G^2 \sum_{t=0}^{E-1}(\eta_t^2(1 + 2\eta_t^2L^2))^t$, $\Upsilon = \prod_{t=0}^{E-1}(1 - 2\eta_t L)^t \frac{2G^2L}{\mu^2} + \frac{LG^2}{2}\sum_{t=0}^{E-1}(1 - 2\eta_t L)^t \eta_t^2$, and $F_c(\mathbf{w}) \triangleq \mathbb{E}_{z \in \mathcal{D}_c}[l(\mathbf{w}, z)]$ denotes the generalization error induced when the model $\mathbf{w}$ is tested at the dataset $\mathcal{D}_c$.*

Theorem 4.3 postulates that the upper bound of the generalization error gap is intrinsically associated with the degrees of non-iid $\delta_k^2$ and $\delta_{k'}^2$. Inspired by this, we propose to utilize the generalization error gap between two agents for the incentive design of the scoring function.

## 5 EQUILIBRIUM CHARACTERIZATION

The preceding findings (Lemma 4.2 & Theorem 4.3) provide insights for the incentive design of the mechanism, which endeavors to minimize the Wasserstein distance, thereby expediting the convergence process artificially. These observations motivate us to formulate our utility/payoff functions, and present a two-stage Stackelberg game model that formalizes the incentive process and examines the existence of equilibrium in federated learning.

### 5.1 PAYMENT DESIGN

**Agents' Utilities.** Leveraging the idea of peer prediction mechanisms [1] (Shnayder et al., 2016; Liu & Wei, 2020), we aim to evaluate the contribution of agents by quantifying the performance disparity (i.e., generalization error gap) between agent $k$ and a randomly selected peer agent $k'$. To achieve this goal, we further develop the scoring rules (*a.k.a.*, payment functions). Note that the upper bound of the generalization error gap can be found in Theorem 4.3, we define the payment function as follows.

$$\text{Payment}_k(e_k) \triangleq f\left(\frac{Q}{\Phi\delta_k^2(e_k) + \Phi\delta_{k'}^2(e_{k'}) + \Upsilon}\right) \propto \frac{1}{\text{Upper\_Bound}(F_c(\mathbf{w}^k) - F_c(\mathbf{w}^{k'}))} \tag{3}$$

where $Q$ denotes the coefficient determined by the learner, *a.k.a.*, initial payment. Basically, the payment function $f : \mathbb{R}^+ \to \mathbb{R}^+$ can be a monotonically non-decreasing function and inversely proportional to $\text{Upper\_Bound}(F_c(\mathbf{w}^k) - F_c(\mathbf{w}^{k'}))$, which indicates that the smaller generalization error gap between one and randomly picked peer agent, the more reward. To facilitate the assessment of the effort level contributed by agents, we rewrite the payment function as $f(e_k, e_{k'})$.

For the cost function, assume each agent $k$ has a marginal cost $c_k$ for producing data samples, the cost for producing data samples to narrow down the non-iid degree of their local data is defined as:

$$\text{Cost}_k(e_k) \triangleq c_k \cdot d(|\delta_k(0) - \delta_k(e_k)|), \forall k \in [N],$$

where $\delta_k(0)$ and $\delta_k(e_k)$ denotes the Wasserstein distance under effort coefficient $e_k = 0$ and $e_k$, respectively. $d(\cdot)$ function scales the distance to reflect the size of collected samples, which would potentially return a larger value when $\delta_k(e_k)$ is close to $\delta_k(0)$. Note that $\delta_k(0)$ signifies the initial value that reflects the original heterogeneous level acting as a constant. For example, in relation to the data-collection scenario (Blum et al., 2021; Karimireddy et al., 2022), one can think that agent $k$ does not own any data samples yet, and therein $\delta_k(0) = 1/2$.

**Discussing why only adding examples lessens non-iid degree.** The influence of training data size is hinted in the convergence bound (Example 1) through a common convergence assumption (Assumption 3.3). With fewer samples, expected local variance bounds increase, harming model performance. Thus, the learner can evaluate the value of $\delta_k$ and adjust the coefficient $Q$ in the payment function, preventing agents from reducing majority class data samples to attain IID.

**The Learner's Payoff.** Given the convergence result shown in Lemma 4.2 and Example 1, we can exactly define the payoff function of the learner as follows.

$$\text{Payoff} \triangleq g\left(\frac{1}{\text{Bound}(\boldsymbol{e})}\right) - T\sum_{k=1}^{N} f(e_k, e_{k'}).$$

where $\text{Bound}(\boldsymbol{e})$ denotes the model's performance upper bound under agents' efforts $\boldsymbol{e}$ and function $g : \mathbb{R}^+ \to \mathbb{R}^+$ can be a monotonically non-decreasing function and inversely proportional to

---

[1]Peer prediction aims to elicit truthful information from individuals without ground-truth verification, by leveraging peer responses assigned for the same task.

$\mathrm{Bound}(\boldsymbol{e})$ representing that the tighter upper bound, the more reward. $T$ indicates the number of communication rounds if we recalculate and reward agents per round. Notice that $\mathrm{Bound}(\boldsymbol{e})$ can be simplified as $\Omega + \gamma \sum_{k=1}^{N} p_k \delta_k^2(e_k)$, and $\Omega$ can be regarded as a constant from Example 1.

## 5.2 TWO-STAGE GAME

The interactions between agents and the learner can be formalized as a two-stage Stackelberg game [2]. In Stage I, the learner strategically determines the reward policy (payment coefficient $Q$) to maximize its payoff. In Stage II, each agent $k$ selects its effort level $e_k$ to maximize its own utility. To obtain an equilibrium, we employ the method of backward induction.

**Agents' Best Responses.** For any agent $k$, he will make efforts with effort level $e_k \neq 0$ if $u(e_k, e_{k'}) \geq u(0, e_{k'})$. This motivates the optimal effort level for the agent. Before diving into finding the existence of equilibrium, we make a mild assumption on payment function $f(\cdot)$:

**Assumption 5.1.** *The payment function $f(\cdot)$ is non-decreasing and strictly concave on $\boldsymbol{e} \in [0, 1]$.*

This assumption states that the speed of obtaining a higher payment is decreasing. It is worth noting that this assumption is natural, reasonable, and aligns with the Ninety-ninety rule. Several commonly used functions, such as the logarithmic function, conform to this mild assumption.

Let us consider a hypothetical scenario where an image classification problem entails the classification of ten classes. In such a case, the agent $k$'s dataset is only sampled from one specific class. It is evident that the Wasserstein distance $\delta_k$ decreases rapidly at the initial stages if agent $k$ invests the same effort. Given the inherent nature of this phenomenon, we assume that it is a well-known characteristic of the payment function for both agents and the learner.

**Theorem 5.1.** *(Optimal effort level). Consider agent $k$ with its marginal cost and the payment function inversely proportional to the generalization error gap with any randomly selected peer $k'$. Then, agent $k$'s optimal effort level $e_k^*$ is:*

$$e_k^* = \begin{cases} 0, & \text{if } \max_{e_k \in [0,1]} u_k(e_k, e_{k'}) \leq 0; \\ \hat{e}_k & \text{such that } \partial f(\hat{e}_k, e_{k'})/\partial \delta_k(\hat{e}_k) + c_k d'(\delta_k(\hat{e}_k)) = 0, & \text{otherwise.} \end{cases}$$

The subsequent section will demonstrate that the establishment of an equilibrium in this two-stage game is contingent upon the specific nature of the problem's configuration. Specifically, we shall prove that an equilibrium solution prevails when the impact of unilateral deviations in an agent's effort level remains limited to their own utility. On the other hand, an equilibrium solution may not exist if infinitesimally small changes to an agent's effort level have an outsized effort on other agents' utilities. The characteristics are formalized into the following definition. Given this definition, we will show the existence of an equilibrium when agents' utility functions are well-behaved (Blum et al., 2021).

**Definition 5.1.** *(Well-behaved utility functions). A set of utility functions $\{u_k : \boldsymbol{e} \to \mathbb{R}^+ | k \in [N]\}$ is considered to be well-behaved over a convex and compact product set $\prod_{k \in [N]}[0, 1] \subseteq \mathbb{R}^N$, if and for each agent $k \in [N]$, there are some constants $d_k^1 \geq 0$ and $d_k^2 \geq 0$ such that for any $\boldsymbol{e} \in \prod_{k \in [N]}[0, 1]$, for all $k' \in [N]$, and $k' \neq k$, $0 \leq \partial u_k(\boldsymbol{e})/\partial e_{k'} \leq d_{k'}^1$, and $\partial u_k(\boldsymbol{e})/\partial e_k \geq d_k^2$.*

**Theorem 5.2.** *(Existence of pure Nash equilibrium). Denote by $\boldsymbol{e}^*$ the optimal effort level, if the utility functions $u_k(\cdot)$s are well-behaved over the set $\prod_{k \in [N]}[0, 1]$, then there exists a pure Nash equilibrium in effort level $\boldsymbol{e}^*$ which for any agent $k$ satisfies,*

$$u_k(e_k^*, \boldsymbol{e}_{-k}^*) \geq u_k(e_k, \boldsymbol{e}_{-k}^*), \quad \forall e_k \in [0, 1], \forall k \in [N].$$

Further discussions on other potential equilibriums and the consistency of extending to all agents can be found in Appendix B.3.

**The Learner's Best Response.** We will solve the learner's problem shown in Problem 2 in Stage I. Given the optimal effort level $\boldsymbol{e}^*$, the learner only needs to calculate the actual initial payment $Q$ for the payment function. In that case, the payoff can be expressed as $\mathrm{Payoff} = g\left(\frac{1}{\mathrm{Bound}(\boldsymbol{e}^*)}\right) - T \sum_{k=1}^{N} f(e_k^*, \boldsymbol{e}_{-k}^*)$. Combined into Problem 2, the optimal solution can be easily computed using a convex program, for example, the standard solver SLSQP in SciPy (Virtanen et al., 2020).

---

[2]Essentially, in the Stackelberg game, players act in sequence; a leader first makes a move, anticipating follower reactions, and then followers optimize their actions based on the leader's decision

## 6 EXPERIMENTS

In this section, we demonstrate the existing heterogeneous effort of non-iid degrees and the performance disparity between agents. We then introduce a logarithmic scoring function to establish an effective incentive mechanism. This mechanism's efficacy is rigorously tested, emphasizing its role in maintaining an equilibrium where no agent can decrease their effort to achieve a higher utility.

### 6.1 EXPERIMENTAL SETUP

**Description.** We adopt the widely utilized federated averaging algorithm (FedAvg) (McMahan et al., 2017) to illustrate the impact of non-iid degrees on effort and incentive issues. We evaluate the performances of several CNN models on four class-balanced image classification datasets: MNIST (LeCun et al., 1998), FashionMNIST (Xiao et al., 2017), CIFAR-10 (Krizhevsky et al., 2009), and CIFAR-100, where the first three are all ten-class image classification tasks. As for CIFAR-100, we employ 20 coarse labels due to an insufficient amount of data in each fine label. Similar to (McMahan et al., 2017), we randomly pick a fixed number of data samples (i.e., 500 or 600) from the whole training dataset, for every round that agents used to train models. To simulate the statistical heterogeneity, we partitioned the data based on the image classes each agent owns. Given $N$ agents who fully participate in the training process, the agent will continuously re-sample the data points with the fixed sample size for each round. For each agent, a proportion of $\delta$ data samples is from a differed majority class, while the remaining samples are uniformly distributed across the rest of the classes. In this way, we can leverage this metric $\delta$ to measure the degree of non-iid qualitatively, while also capturing the idea of Wasserstein distance. Due to space limits, we defer more details to Appendix C.1, i.e., data split, parameter settings, etc.

### 6.2 PERFORMANCE RESULTS ON NON-IID DEGREES

**Heterogeneous efforts.** Figure 1 demonstrates the performances of three instantize distributions with different $\delta_k$s on four datasets, respectively. In Figure 1, it is apparent that the degree of non-iid has a substantial impact on the actual performance. The greater heterogeneity in the data, the worse the performance. Specifically, $\delta = 0.1$ corresponds to the iid setting for MNIST, FashionMNIST, and CIFAR-10 datasets, when we equally distribute the remaining samples into other classes.

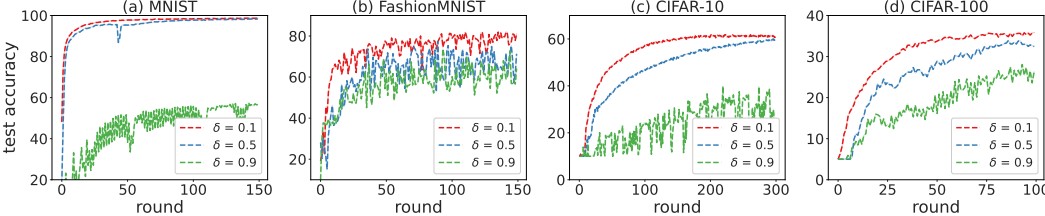

Figure 1: FL training process under different non-iid degrees.

**Performance comparison with peers.** We depict the test accuracy of the local models from two agents with different fixed non-iid degrees to show the performance gap between them caused by statistical heterogeneity. In particular, we set the non-iid degree of two agents as $0.1$ and $0.9$, respectively. Figure 2 illustrates the existing performance gap between two agents. It is evident that the presence of the performance gap can be attributed to the heterogeneous efforts of the data, thereby indicating the feasibility of utilizing it as an effective measure for designing payment functions. Due to space limit, additional results with other non-iid metrics or settings can be found in Appendix C.2.

### 6.3 IMPLEMENTING INCENTIVE MECHANISM

**Scoring function.** We first present a scoring structure as the payment function $f(\cdot)$: a logarithmic function with natural base $e$. For simplicity, the cost functions are designed as classical linear functions. For more details, please refer to Appendix C.3. We can derive and obtain the optimal effort level for logarithmic functions, which can be generally rewritten into a general form $\delta_k \triangleq f(\delta_{k'}^2) = \frac{1}{c_k} \pm \sqrt{\frac{1}{c_k^2} - \delta_{k'}^2 - \frac{\Upsilon}{\Phi}}$, where $\delta_{k'}$ is the non-iid degree of a randomly selected peer.

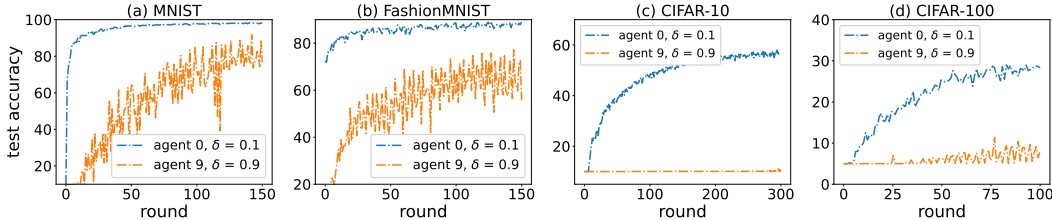

Figure 2: Performance comparison with peers. In default, the non-iid degree is 0.5.

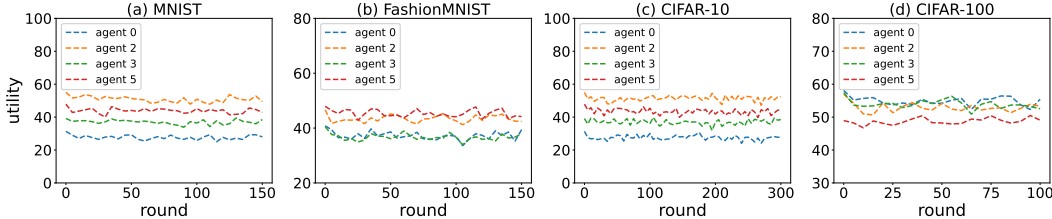

Figure 3: Utility variation using a logarithmic function.

**Parameter analysis.** Combined with Theorem 4.3, payment function mainly depends on two parameters $\Phi$ and $\Upsilon$, which both depend on learning rate $\eta$, Lipschitz constant $L$ and gradient bound $G$. Recall that we set the learning rate $\eta = 0.01$, and remain constant during training, *i.e.*, $\eta_t = \eta, \forall t$. The detailed analysis of Lipschitz constant $L$ and gradient bound $G$ are presented in Appendix C.3. We can obtain that $\Phi \in [300, 30000]$ and $\Upsilon \in [200, 20000]$, respectively. Due to the Ninety-ninety rule, we set the effort-distance function $\delta_k(e_k)$ as an exponential function here: $\delta_k(e_k) = \exp(-e_k)$.

**Existing equilibrium.** This part is to illustrate the existing equilibrium that no one can unilaterally change his invested effort level to obtain a higher utility. As shown in Figure 3, the utilities of four randomly selected agents remain stable, which indicates an equilibrium. Note that the existing fluctuation of utility mainly results from the randomness of selecting peers.

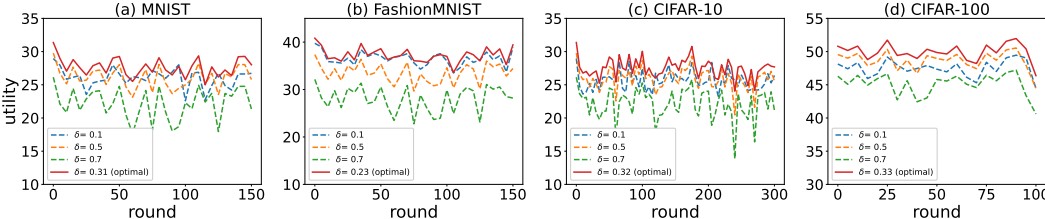

Figure 4: Utility variation using a logarithmic function under different non-iid degrees.

**Impact on utility under different non-iid levels.** We further give a detailed example in Figure 4, to illustrate the impact shown in utility if anyone unilaterally increases or decreases his invested effort level. More specifically, we present the possible results of a specific single client when he changes his invested effort levels, however, other clients hold their optimal invested effort levels. To mitigate the impact of randomness, we calculate the average values of the utilities within 5 communication rounds. Even though there is much fluctuation brought by the randomness of selecting peers, the utility achieved by the optimal effort levels is much larger than others in expectation.

## 7 CONCLUSION

In this paper, we utilize Wasserstein distance to quantify the non-iid degree of an agent's local training data, and are the first to prove its significance in the convergence bound of FL with heterogeneous data. Unlike recent game-theoretic methods centered on data-sharing scenarios, our incentive-aware mechanism promotes faster convergence by accounting for data heterogeneity and ensures truthful reporting from agents. We also explore the equilibrium presence in our framework. Experiments on real-world datasets validate the efficacy of our incentive mechanism.

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

## APPENDIX

**Broader Impacts.**   Incentive mechanisms in FL frequently encounter challenges due to uninformative and low-quality responses from agents. Our work aims to encourage agents to respond truthfully, simultaneously pursuing high-quality data to accelerate the process of convergence. Our proposed incentive-aware framework yields valuable insights into this research, and we anticipate that its influence could extend even more broadly. We believe that our framework will attract the attention of machine learning professionals and researchers keen on developing incentive mechanisms for strategic federated learning.

**Code of Ethics.**   There is no sensitive attribute in our method and results. Therefore, the potential negative impacts do not apply to our work. We will open-source the code when the paper is published.

**Limitations.**   While our study offers valuable insights, it does have certain limitations. For example, the adopted four well-recognized datasets are all image datasets. Our future work includes exploring the effectiveness of our proposed incentive mechanism on text datasets.

## ORGANIZATION OF THE APPENDIX

In this supplementary material, we provide missing proofs for all theoretical results in this paper. The Appendix is organized as follows.

- Section A presents annotation and omitted proofs in Section 4, including Lemma 4.2 and Theorem 4.3. In particular, we provide more examples in A.6 to illustrate the broad applicability of our Lemma 4.2 to existing convergence results. More discussions about the assumption of bounding global gradient variance are also presented in Section A.7.
- Section B provides more discussions about Assumption 5.1, omitted proofs in Section 5 (Theorem 5.1 and Theorem 5.2), and more discussion about the existence of other possible equilibrium and the consistency of extending to all agents.
- Section C presents experiment details and additional results, including parameter settings, scoring function design, and gradient divergence.

## A   OMITTED PROOFS IN SECTION 4

Before presenting the proofs, we first introduce some annotations in Section A.1.

### A.1   ANNOTATION

Here, we rewrite the expected loss over $k$-th agent's dataset $\mathcal{D}_k$ by splitting samples according to their labels, shown as follows.

$$F_k(\mathbf{w}) \triangleq \mathbb{E}_{\zeta^k \in \mathcal{D}_k}[l(\mathbf{w}, \zeta^k)] = \mathbb{E}_{(\boldsymbol{x}, y) \sim p^{(k)}}\left[\sum_{i=1}^{I} \ell_i(\mathbf{w}, \boldsymbol{x})\right] = \sum_{i=1}^{I} p^{(k)}(y=i)\mathbb{E}_{\boldsymbol{x}|y=i}[\ell(\mathbf{w}, \boldsymbol{x})].$$

For each round (epoch) $t$ on agent $k$, its local optimizer performs SGD as the following:

$$\mathbf{w}_t^k = \mathbf{w}_{t-1}^k - \eta_t \sum_{i=1}^{I} p^{(k)}(y=i)\nabla_{\mathbf{w}}\mathbb{E}_{\boldsymbol{x}|y=i}[\ell(\mathbf{w}_{t-1}^k, \boldsymbol{x})].$$

Let's assume that the learner aggregates local updates from agents after every $E$ step. For instance, the aggregated weight at the $m$-th synchronization is represented by $\overline{\mathbf{w}}_{mE}$. Here, for analysis, we introduce a virtual aggregated sequence $\overline{\mathbf{w}}_t = \sum_{k=1}^{N} p_k \mathbf{w}_t^k$.

The discrepancy between the iterates from agents' average $\overline{\mathbf{w}}_t$ over a single iteration is quantified using the term $||\overline{\mathbf{w}}_t - \mathbf{w}_t^k||$. Here, Lemma 4.2 shows the upper bound of $||\overline{\mathbf{w}}_t - \mathbf{w}_t^k||$ according to $E$ synchronization interval.

### A.2   PROOF OF LEMMA 4.2

**Lemma 4.2** (Bounded the divergence of $\{\mathbf{w}_t^k\}$). Let Assumption 3.4 hold and $G$ be defined therein, given the synchronization interval $E$ (local epochs), the learning rate $\eta_t$. Suppose that $\nabla_{\mathbf{w}}\mathbb{E}_{\boldsymbol{x}|y=i}[\ell_i(\boldsymbol{x}, \mathbf{w})]$ is $L_{\boldsymbol{x}|y=i}$-

Lipschitz, it follows that

$$\mathbb{E}[\sum_{k=1}^{N} p_k ||\overline{\mathbf{w}}_t - \mathbf{w}_t^k||^2] \le 16(E-1)G^2 \eta_t^2 (1+2\eta_t)^{2(E-1)} \sum_{k=1}^{N} p_k \underbrace{\left( \sum_{i=1}^{I} |p^{(k)}(y=i) - p^{(c)}(y=i)| \right)^2}_{4\delta_k^2}$$

*Proof.* Without loss of generality, suppose that $mE \le t < (m+1)E$, where $m \in \mathbb{Z}^+$. Given the definitions of $\overline{\mathbf{w}}_t$ and $\mathbf{w}_t^k$ , we have

$$||\overline{\mathbf{w}}_t - \mathbf{w}_t^k||$$

$$= ||\sum_{k'=1}^{N} p_{k'} \mathbf{w}_t^{k'} - \mathbf{w}_t^k||$$

$$= ||\sum_{k'=1}^{N} p_{k'} (\mathbf{w}_{t-1}^{k'} - \eta_t \sum_{i=1}^{I} p^{(k')}(y=i) \nabla_{\mathbf{w}} \mathbb{E}_{\boldsymbol{x}|y=i}[\ell(\mathbf{w}_{t-1}^{k'}, \boldsymbol{x})]) - \mathbf{w}_{t-1}^k + \eta_t \sum_{i=1}^{I} p^{(k)}(y=i) \nabla_{\mathbf{w}} \mathbb{E}_{\boldsymbol{x}|y=i}[\ell(\mathbf{w}_{t-1}^k, \boldsymbol{x})]||$$

$$\le ||\sum_{k'=1}^{N} p_{k'} \mathbf{w}_{t-1}^{k'} - \mathbf{w}_{t-1}^k||$$

$$+ \eta_t ||\sum_{k'=1}^{N} p_{k'} \sum_{i=1}^{I} p^{(k')}(y=i) \nabla_{\mathbf{w}} \mathbb{E}_{\boldsymbol{x}|y=i}[\ell(\mathbf{w}_{t-1}^{k'}, \boldsymbol{x})] - \sum_{i=1}^{I} p^{(k)}(y=i) \nabla_{\mathbf{w}} \mathbb{E}_{\boldsymbol{x}|y=i}[\ell(\mathbf{w}_{t-1}^k, \boldsymbol{x})]||$$

(from triangle inequality)

$$\le ||\sum_{k'=1}^{N} p_{k'} \mathbf{w}_{t-1}^{k'} - \mathbf{w}_{t-1}^k||$$

$$+ \eta_t \sum_{k'=1}^{N} p_{k'} ||\sum_{i=1}^{I} p^{(k')}(y=i) \nabla_{\mathbf{w}} \mathbb{E}_{\boldsymbol{x}|y=i}[\ell(\mathbf{w}_{t-1}^{k'}, \boldsymbol{x})] - \sum_{i=1}^{I} p^{(k)}(y=i) \nabla_{\mathbf{w}} \mathbb{E}_{\boldsymbol{x}|y=i}[\ell(\mathbf{w}_{t-1}^k, \boldsymbol{x})]||$$

$$\le \sum_{k'=1}^{N} p_{k'} ||\mathbf{w}_{t-1}^{k'} - \mathbf{w}_{t-1}^k||$$

$$+ \eta_t \sum_{k'=1}^{N} p_{k'} ||\sum_{i=1}^{I} p^{(k')}(y=i) \cdot [\nabla_{\mathbf{w}} \mathbb{E}_{\boldsymbol{x}|y=i}[\ell(\mathbf{w}_{t-1}^{k'}, \boldsymbol{x})] - \nabla_{\mathbf{w}} \mathbb{E}_{\boldsymbol{x}|y=i}[\ell(\mathbf{w}_{t-1}^k, \boldsymbol{x})]]||$$

$$+ \eta_t \sum_{k'=1}^{N} p_{k'} ||\sum_{i=1}^{I} [p^{(k')}(y=i) - p^{(k)}(y=i)] \cdot \nabla_{\mathbf{w}} \mathbb{E}_{\boldsymbol{x}|y=i}[\ell(\mathbf{w}_{t-1}^k, \boldsymbol{x})]||$$

$$\le \sum_{k'=1}^{N} p_{k'} (1 + \eta_t \sum_{i=1}^{I} p^{(k')}(y=i) L_{\boldsymbol{x}|y=i}) ||\mathbf{w}_{t-1}^{k'} - \mathbf{w}_{t-1}^k||$$

$$+ \eta_t \sum_{k'=1}^{N} p_{k'} g_{max}(\mathbf{w}_{t-1}^k) \sum_{i=1}^{I} |p^{(k')}(y=i) - p^{(k)}(y=i)|.$$

Recall that $p_k$ indicates the weight of the $k$-th agent such that $p_k \ge 0$ and $\sum_{k=1}^{N} p_k = 1$, which follows FedAvg (McMahan et al., 2017). The last inequality holds because we assume that $\nabla_{\mathbf{w}} \mathbb{E}_{\boldsymbol{x}|y=i}[\ell(\mathbf{w}_t^k, \boldsymbol{x})]$ is $L_{\boldsymbol{x}|y=i}$-Lipschitz, *i.e.*, $||\nabla_{\mathbf{w}} \mathbb{E}_{\boldsymbol{x}|y=i}[\ell(\mathbf{w}_t^{k'}, \boldsymbol{x})] - \nabla_{\mathbf{w}} \mathbb{E}_{\boldsymbol{x}|y=i}[\ell(\mathbf{w}_t^k, \boldsymbol{x})]|| \le L_{\boldsymbol{x}|y=i} ||\mathbf{w}_t^{k'} - \mathbf{w}_t^k||$, and denote $g_{max}(\mathbf{w}_t^k) = \max_{i=1}^{I} ||\nabla_{\mathbf{w}} \mathbb{E}_{\boldsymbol{x}|y=i}[\ell(\mathbf{w}_t^k, \boldsymbol{x})]||$.

Subsequently, the term $||\mathbf{w}_{t-1}^{k'} - \mathbf{w}_{t-1}^k||$ illustrated in the above inequality can be further deduced as follows.

$$||\mathbf{w}_{t-1}^{k'} - \mathbf{w}_{t-1}^k||$$

$$= ||\mathbf{w}_{t-2}^{k'} - \eta_{t-1}\sum_{i=1}^{I} p^{(k')}(y=i)\nabla_{\mathbf{w}}\mathbb{E}_{\boldsymbol{x}|y=i}[\ell(\mathbf{w}_{t-2}^{k'}, \boldsymbol{x})] - \mathbf{w}_{t-2}^k + \eta_{t-1}\sum_{i=1}^{I} p^{(k)}(y=i)\nabla_{\mathbf{w}}\mathbb{E}_{\boldsymbol{x}|y=i}[\ell(\mathbf{w}_{t-2}^k, \boldsymbol{x})]||$$

$$\leq ||\mathbf{w}_{t-2}^{k'} - \mathbf{w}_{t-2}^k||$$

$$+ \eta_{t-1}||\sum_{i=1}^{I} p^{(k')}(y=i)\nabla_{\mathbf{w}}\mathbb{E}_{\boldsymbol{x}|y=i}[\ell(\mathbf{w}_{t-2}^{k'}, \boldsymbol{x})] - \eta_{t-1}\sum_{i=1}^{I} p^{(k)}(y=i)\nabla_{\mathbf{w}}\mathbb{E}_{\boldsymbol{x}|y=i}[\ell(\mathbf{w}_{t-2}^k, \boldsymbol{x})]||$$

$$\leq ||\mathbf{w}_{t-2}^{k'} - \mathbf{w}_{t-2}^k|| + \eta_{t-1}\sum_{i=1}^{I} p^{(k')}(y=i)||\nabla_{\mathbf{w}}\mathbb{E}_{\boldsymbol{x}|y=i}[\ell(\mathbf{w}_{t-2}^{k'}, \boldsymbol{x})] - \nabla_{\mathbf{w}}\mathbb{E}_{\boldsymbol{x}|y=i}[\ell(\mathbf{w}_{t-2}^k, \boldsymbol{x})]||$$

$$+ \eta_{t-1}g_{max}(\mathbf{w}_{t-2}^k)\sum_{i=1}^{I} |p^{(k')}(y=i) - p^{(k)}(y=i)|$$

$$\leq (1 + \eta_{t-1}\sum_{i=1}^{I} p^{(k')}(y=i)L_{\boldsymbol{x}|y=i})||\mathbf{w}_{t-2}^{k'} - \mathbf{w}_{t-2}^k|| + \eta_{t-1}g_{max}(\mathbf{w}_{t-2}^k)\sum_{i=1}^{I} |p^{(k')}(y=i) - p^{(k)}(y=i)|.$$

Here, we make a mild assumption that $L = L_{\boldsymbol{x}|y=i} = L_{\boldsymbol{x}|y=i'}, \forall i, i'$, implying that the Lipschitz-continuity remains the same regardless of the classes of the samples. We denote $\alpha_t = 1 + \eta_t L$ for simplicity. Plugging $||\mathbf{w}_{t-1}^{k'} - \mathbf{w}_{t-1}^k||$ into the expression of $||\overline{\mathbf{w}}_t - \mathbf{w}_t^k||$, by induction, we have,

$$||\overline{\mathbf{w}}_t - \mathbf{w}_t^k||$$

$$\leq \sum_{k'=1}^{N} p_{k'}\alpha_t||\mathbf{w}_{t-1}^{k'} - \mathbf{w}_{t-1}^k|| + \eta_t\sum_{k'=1}^{N} p_{k'}g_{max}(\mathbf{w}_{t-1}^k)\sum_{i=1}^{I} |p^{(k')}(y=i) - p^{(k)}(y=i)|$$

$$\leq \sum_{k'=1}^{N} p_{k'}\alpha_t\alpha_{t-1}||\mathbf{w}_{t-2}^{k'} - \mathbf{w}_{t-2}^k|| + \eta_t\sum_{k'=1}^{N} p_{k'}g_{max}(\mathbf{w}_{t-1}^k)\sum_{i=1}^{I} |p^{(k')}(y=i) - p^{(k)}(y=i)|$$

$$+ \eta_{t-1}\sum_{k'=1}^{N} p_{k'}\alpha_t g_{max}(\mathbf{w}_{t-2}^k)\sum_{i=1}^{I} |p^{(k')}(y=i) - p^{(k)}(y=i)|$$

$$\leq \sum_{k'=1}^{N} p_{k'}\prod_{t'=t_0}^{t-1}\alpha_{t'+1}||\mathbf{w}_{t_0}^{k'} - \mathbf{w}_{t_0}^k||$$

$$+ \sum_{k'=1}^{N} p_{k'}\sum_{t'=t_0}^{t-1}\eta_{t'+1}\prod_{t''=t'+1}^{t-1}\alpha_{t''+1}g_{max}(\mathbf{w}_{t'}^k)\sum_{i=1}^{I} |p^{(k')}(y=i) - p^{(k)}(y=i)|.$$

Recall that $mE \leq t < (m+1)E$. When $t_0 = t - t' = mE$, for any $k', k \in \mathcal{N}, \overline{\mathbf{w}}_{t_0} = \mathbf{w}_{t_0}^{k'} = \mathbf{w}_{t_0}^k$. In this case, the first term of the last inequality $\sum_{k'=1}^{N} p_{k'}\prod_{t'=t_0}^{t-1}\alpha_{t'+1}||\mathbf{w}_{t'}^{k'} - \mathbf{w}_{t'}^k||$ equals 0. Then, we have,

$$\mathbb{E}[\sum_{k=1}^{N} p_k||\overline{\mathbf{w}}_t - \mathbf{w}_t^k||^2]$$

$$= \sum_{k=1}^{N} p_k\mathbb{E}[||\overline{\mathbf{w}}_t - \mathbf{w}_t^k||^2]$$

$$\leq \sum_{k=1}^{N} p_k\mathbb{E}[||\sum_{k'=1}^{N} p_{k'}\sum_{i=1}^{I} |p^{(k')}(y=i) - p^{(k)}(y=i)|\sum_{t'=t_0}^{t-1}\eta_{t'+1}\prod_{t''=t'+1}^{t-1}\alpha_{t''+1}g_{max}(\mathbf{w}_{t'}^k)||^2]$$

$$\leq \sum_{k=1}^{N} p_k\sum_{k'=1}^{N} p_{k'}\left(\sum_{i=1}^{I} |p^{(k')}(y=i) - p^{(k)}(y=i)|\right)^2\sum_{t'=t_0}^{t-1}\eta_{t'+1}^2\prod_{t''=t'+1}^{t-1}\alpha_{t''+1}^2\mathbb{E}[||g_{max}(\mathbf{w}_{t'}^k)||^2]$$

$$\leq (E-1)G^2\eta_{t_0}^2\alpha_{t_0}^{2(E-1)}\sum_{k=1}^{N}\sum_{k'=1}^{N} p_k p_{k'}\left(\sum_{i=1}^{I} |p^{(k')}(y=i) - p^{(k)}(y=i)|\right)^2$$

$$\leq 4(E-1)G^2\eta_t^2(1+2\eta_t)^{2(E-1)}\sum_{k=1}^{N}\sum_{k'=1}^{N} p_k p_{k'}\left(\sum_{i=1}^{I} |p^{(k')}(y=i) - p^{(k)}(y=i)|\right)^2.$$

(4)

where in the last inequality, $\mathbb{E}[||g_{max}(\mathbf{w}_{t'}^k)||^2] \leq G^2$ because of Assumption 3.4, and we use $\eta_{t_0} \leq 2\eta_{t_0+E} \leq 2\eta_t$ for $t_0 \leq t \leq t_0 + E$. In order to show the degree of non-iid, we rewrite the last term $\sum_{k=1}^N \sum_{k'=1}^N p_k p_{k'} \left( \sum_{i=1}^I |p^{(k')}(y=i) - p^{(k)}(y=i)| \right)^2$ as follows.

$$\sum_{k=1}^N \sum_{k'=1}^N p_k p_{k'} \left( \sum_{i=1}^I |p^{(k')}(y=i) - p^{(k)}(y=i)| \right)^2$$

$$\leq 2 \sum_{k=1}^N \sum_{k'=1}^N p_k p_{k'} \left( \sum_{i=1}^I |p^{(k')}(y=i) - p^{(c)}(y=i)| \right)^2 + 2 \sum_{k=1}^N \sum_{k'=1}^N p_k p_{k'} \left( \sum_{i=1}^I |p^{(k)}(y=i) - p^{(c)}(y=i)| \right)^2$$

$$= 2 \sum_{k'=1}^N p_{k'} \left( \sum_{i=1}^I |p^{(k')}(y=i) - p^{(c)}(y=i)| \right)^2 + 2 \sum_{k=1}^N p_k \left( \sum_{i=1}^I |p^{(k)}(y=i) - p^{(c)}(y=i)| \right)^2$$

$$= 4 \sum_{k=1}^N p_k \left( \sum_{i=1}^I |p^{(k)}(y=i) - p^{(c)}(y=i)| \right)^2$$

$$= 16 \sum_{k=1}^N p_k \delta_k^2$$

$$\tag{5}$$

where in the first inequality, we use the fact that $||x+y||^2 \leq 2||x||^2 + 2||y||^2$, and $p^{(c)}$ represents the actual reference data distribution in the centralized setting.

Therefore, substituting the term in Eq. (5) into Eq. (4), we have,

$$\mathbb{E}[\sum_{k=1}^N p_k ||\overline{\mathbf{w}}_t - \mathbf{w}_t^k||^2] \leq 64(E-1)G^2 \eta_t^2 (1+2\eta_t)^{2(E-1)} \sum_{k=1}^N p_k \delta_k^2.$$

$\square$

## A.3 PROOF OF THEOREM 4.3

**Theorem 4.3** Suppose that the expected loss function $F_c(\cdot)$ also follows Assumption 3.2, then the upper bound of the generalization error gap between agent $k$ and $k'$ is

$$F_c(\mathbf{w}^k) - F_c(\mathbf{w}^{k'}) \leq \Phi \delta_k^2(e_k) + \Phi \delta_{k'}^2(e_{k'}) + \Upsilon$$

where $\Phi = 16L^2 G^2 \sum_{t=0}^{E-1} (\eta_t^2(1+2\eta_t^2 L^2))^t$, $\Upsilon = \prod_{t=0}^{E-1} (1-2\eta_t L)^t \frac{2G^2 L}{\mu^2} + \frac{LG^2}{2} \sum_{t=0}^{E-1} (1-2\eta_t L)^t \eta_t^2$, and $F_c(\mathbf{w}) \triangleq \mathbb{E}_{z \in \mathcal{D}_c}[l(\mathbf{w}, z)]$ denotes the generalization error induced when the model $\mathbf{w}$ is tested at the dataset $D_c$.

*Proof.* Due to the L-smoothness, we have

$$F_c(\mathbf{w}^k) - F_c(\mathbf{w}^{k'}) \leq \langle \nabla F_c(\mathbf{w}^{k'}), \mathbf{w}^k - \mathbf{w}^{k'} \rangle + \frac{L}{2} \|\mathbf{w}^k - \mathbf{w}^{k'}\|^2.$$

By Cauchy-Schwarz inequality and AM-GM inequality, we have

$$\langle \nabla F_c(\mathbf{w}^{k'}), \mathbf{w}^k - \mathbf{w}^{k'} \rangle \leq \frac{L}{2} \|\mathbf{w}^k - \mathbf{w}^{k'}\|^2 + \frac{1}{2L} \|\nabla F_c(\mathbf{w}^{k'})\|^2.$$

Then, due to the L-smoothness of $F_c(\cdot)$ (Assumption 3.2), we can get a variant of Polak-Łojasiewicz inequality, which follows

$$\|\nabla F_c(\mathbf{w}^{k'})\|^2 \leq 2L(F_c(\mathbf{w}^{k'}) - F_c^*) \leq L^2 \|\mathbf{w}^{k'} - \mathbf{w}^*\|^2.$$

Therefore, we have

$$F_c(\mathbf{w}^k) - F_c(\mathbf{w}^{k'}) \leq L\|\mathbf{w}^k - \mathbf{w}^{k'}\|^2 + \frac{L}{2}\|\mathbf{w}^{k'} - \mathbf{w}^*\|^2.$$

Combined with Lemma A.1 and A.2, we finished the proof. $\square$

## A.4 PROOF OF LEMMA A.1

**Lemma A.1.** *Suppose Assumptions 3.2 to 3.3 hold, then we have*

$$\|\mathbf{w}^k - \mathbf{w}^{k'}\|^2 \le 16LG^2 \sum_{t=0}^{E-1} (\eta_t^2(1 + 2\eta_t^2 L^2))^t \left( \delta_k^2 + \delta_{k'}^2 \right).$$

*Proof.* For any time-step $t+1$, we have

$$\|\mathbf{w}_{t+1}^k - \mathbf{w}_{t+1}^{k'}\|^2$$

$$= \|\mathbf{w}_t^k - \eta_t \sum_{i=1}^I p^{(k)}(y = i) \nabla_{\mathbf{w}} \mathbb{E}_{\mathbf{x}|y=i}[\ell(\mathbf{w}_t^k, \mathbf{x})] - \mathbf{w}_t^{k'} + \eta_t \sum_{i=1}^I p^{(k')}(y = i) \nabla_{\mathbf{w}} \mathbb{E}_{\mathbf{x}|y=i}[\ell(\mathbf{w}_t^{k'}, \mathbf{x})]\|^2$$

$$\le \|\mathbf{w}_t^k - \mathbf{w}_t^{k'}\|^2 + \eta_t^2 \|\sum_{i=1}^I p^{(k)}(y = i) \nabla_{\mathbf{w}} \mathbb{E}_{\mathbf{x}|y=i}[\ell(\mathbf{w}_t^k, \mathbf{x})] - \sum_{i=1}^I p^{(k')}(y = i) \nabla_{\mathbf{w}} \mathbb{E}_{\mathbf{x}|y=i}[\ell(\mathbf{w}_t^{k'}, \mathbf{x})]\|^2$$

$$\le \|\mathbf{w}_t^k - \mathbf{w}_t^{k'}\|^2 + 2\eta_t^2 \|\sum_{i=1}^I p^{(k')}(y = i) L_{\mathbf{x}|y=i} \left[ \nabla_{\mathbf{w}} \mathbb{E}_{\mathbf{x}|y=i}[\ell(\mathbf{w}_t^k, \mathbf{x})] - \nabla_{\mathbf{w}} \mathbb{E}_{\mathbf{x}|y=i}[\ell(\mathbf{w}_t^{k'}, \mathbf{x})] \right]\|^2$$

$$+ 2\eta_t^2 \|\sum_{i=1}^I \left( p^{(k)}(y = i) - p^{(k')}(y = i) \right) \nabla_{\mathbf{w}} \mathbb{E}_{\mathbf{x}|y=i}[\ell(\mathbf{w}_t^{k'}, \mathbf{x})]\|^2$$

$$\le \|\mathbf{w}_t^k - \mathbf{w}_t^{k'}\|^2 + 2\eta_t^2 \left( \sum_{i=1}^I p^{(k)}(y = i) L_{\mathbf{x}|y=i} \right)^2 \|\mathbf{w}_t^k - \mathbf{w}_t^{k'}\|^2$$

$$+ 2L\eta_t^2 g_{max}^2(\mathbf{w}_t^{k'}) \left( \sum_{i=1}^I |p^{(k)}(y = i) - p^{(k')}(y = i)| \right)^2$$

$$\le \left( 1 + 2\eta_t^2 \left( \sum_{i=1}^I p^{(k)}(y = i) L_{\mathbf{x}|y=i} \right)^2 \right) \|\mathbf{w}_t^k - \mathbf{w}_t^{k'}\|^2$$

$$+ 2L\eta_t^2 g_{max}^2(\mathbf{w}_t^{k'}) \left( \sum_{i=1}^I |p^{(k)}(y = i) - p^{(k')}(y = i)| \right)^2$$

$$\le (1 + 2\eta_t^2 L^2) \|\mathbf{w}_t^k - \mathbf{w}_t^{k'}\|^2 + 2L\eta_t^2 G^2 \left( \sum_{i=1}^I |p^{(k)}(y = i) - p^{(k')}(y = i)| \right)^2.$$

where the third inequality holds because we assume that $\nabla_{\mathbf{w}} \mathbb{E}_{\mathbf{x}|y=i}[\ell(\mathbf{x}, \mathbf{w})]$ is $L_{\mathbf{x}|y=i}$-Lipschitz continuous, *i.e.*, $\|\nabla_{\mathbf{w}} \mathbb{E}_{\mathbf{x}|y=i}[\ell(\mathbf{w}_t^k, \mathbf{x})] - \nabla_{\mathbf{w}} \mathbb{E}_{\mathbf{x}|y=i}[\ell(\mathbf{w}_t^{k'}, \mathbf{x})]\| \le L_{\mathbf{x}|y=i} \|\mathbf{w}_t^k - \mathbf{w}_t^{k'}\|$, and denote $g_{max}(\mathbf{w}_t^{k'}) = max_{k=1}^I \|\nabla_{\mathbf{w}} \mathbb{E}_{\mathbf{x}|y=i}[\ell(\mathbf{w}_t^{k'}, \mathbf{x})]\|$. The last inequality holds because the above-mentioned assumption that $L = L_{\mathbf{x}|y=i} = L_{\mathbf{x}|y=i'}, \forall i, i'$, *i.e.*, Lipschitz-continuity will not be affected by the samples' classes. Then, $g_{max}(\mathbf{w}_t^{k'}) \le G$ because of Assumption 3.4.

When we focus on the aggregation within a single communication round $E$ analogous to the one-shot federated learning setting, we have

$$\|\mathbf{w}_{E-1}^k - \mathbf{w}_{E-1}^{k'}\|^2$$

$$\le (1 + 2\eta_t^2 L^2) \|\mathbf{w}_{E-2}^k - \mathbf{w}_{E-2}^{k'}\|^2 + 2L\eta_t^2 G^2 \left( \sum_{i=1}^I |p^{(k)}(y = i) - p^{(k')}(y = i)| \right)^2$$

$$\le \prod_{t=0}^{E-1} (1 + 2\eta_t^2 L^2)^t \|\mathbf{w}_0^k - \mathbf{w}_0^{k'}\|^2 + 2LG^2 \sum_{t=0}^{E-1} (\eta_t^2(1 + 2\eta_t^2 L^2))^t \left( \sum_{i=1}^I |p^{(k)}(y = i) - p^{(k')}(y = i)| \right)^2$$

$$\le 2LG^2 \sum_{t=0}^{E-1} (\eta_t^2(1 + 2\eta_t^2 L^2))^t \left( \sum_{i=1}^I |p^{(k)}(y = i) - p^{(k')}(y = i)| \right)^2.$$

The last inequality holds because for any agent $k, k' \in [N]$, $\mathbf{w}_0 = \mathbf{w}_0^k = \mathbf{w}_0^{k'}$. In order to obtain the non-iid degree $\delta_k$, similarly, we can rewrite the last term $\left( \sum_{i=1}^I |p^{(k)}(y = i) - p^{(k')}(y = i)| \right)^2$ as follows.

$$\left( \sum_{i=1}^{I} |p^{(k)}(y=i) - p^{(k')}(y=i)| \right)^2$$

$$\leq \left( \sum_{i=1}^{I} |p^{(k)}(y=i) - p^{(c)}(y=k)| + \sum_{i=1}^{I} |p^{(k')}(y=i) - p^{(c)}(y=k)| \right)^2$$

$$\leq 2 \left( \sum_{i=1}^{I} |p^{(k')}(y=i) - p^{(c)}(y=i)| \right)^2 + 2 \left( \sum_{i=1}^{I} |p^{(k)}(y=i) - p^{(c)}(y=i)| \right)^2$$

$$= 8(\delta_k^2 + \delta_{k'}^2).$$

where in the first inequality, we use the fact that $||x+y||^2 \leq 2||x||^2 + 2||y||^2$, and $p^{(c)}$ represents the actual reference data distribution in the centralized setting. □

### A.5 PROOF OF LEMMA A.2

**Lemma A.2.** *Suppose that Assumption 3.2 and Assumption 3.4 hold, then we have*

$$\|\mathbf{w}^{k'} - \mathbf{w}^*\|^2 \leq \prod_{t=0}^{E-1} (1 - 2\eta_t L)^t \frac{4G^2}{\mu^2} + G^2 \sum_{t=0}^{E-1} (1 - 2\eta_t L)^t \eta_t^2.$$

*Proof.* Following the classical idea, we have

$$\mathbb{E}\left[ \left\| \mathbf{w}_{t+1}^{k'} - \mathbf{w}^* \right\|^2 \right]$$

$$= \mathbb{E}\left[ \left\| \Pi_{\mathcal{W}} \left( \mathbf{w}_t^{k'} - \eta_t \hat{\mathbf{g}}_t \right) - \mathbf{w}^* \right\|^2 \right]$$

$$\leq \mathbb{E}\left[ \left\| \mathbf{w}_t^{k'} - \eta_t \hat{\mathbf{g}}_t - \mathbf{w}^* \right\|^2 \right]$$

$$= \mathbb{E}\left[ \left\| \mathbf{w}_t^{k'} - \mathbf{w}^* \right\|^2 \right] - 2\eta_t \mathbb{E}\left[ \left\langle \hat{\mathbf{g}}_t, \mathbf{w}_t^{k'} - \mathbf{w}^* \right\rangle \right] + \eta_t^2 \mathbb{E}\left[ \|\hat{\mathbf{g}}_t\|^2 \right]$$

$$= \mathbb{E}\left[ \left\| \mathbf{w}_t^{k'} - \mathbf{w}^* \right\|^2 \right] - 2\eta_t \mathbb{E}\left[ \left\langle \mathbf{g}_t, \mathbf{w}_t^{k'} - \mathbf{w}^* \right\rangle \right] + \eta_t^2 \mathbb{E}\left[ \|\hat{\mathbf{g}}_t\|^2 \right]$$

$$\leq \mathbb{E}\left[ \left\| \mathbf{w}_t^{k'} - \mathbf{w}^* \right\|^2 \right] - 2\eta_t \mathbb{E}\left[ F\left( \mathbf{w}_t^{k'} \right) - F\left( \mathbf{w}^* \right) + \frac{L}{2} \left\| \mathbf{w}_t^{k'} - \mathbf{w}^* \right\|^2 \right] + \eta_t^2 G^2$$

$$\leq \mathbb{E}\left[ \left\| \mathbf{w}_t^{k'} - \mathbf{w}^* \right\|^2 \right] - 2\eta_t \mathbb{E}\left[ \frac{L}{2} \left\| \mathbf{w}_t^{k'} - \mathbf{w}^* \right\|^2 + \frac{L}{2} \left\| \mathbf{w}_t^{k'} - \mathbf{w}^* \right\|^2 \right] + \eta_t^2 G^2$$

$$= (1 - 2\eta_t L) \mathbb{E}\left[ \left\| \mathbf{w}_t^{k'} - \mathbf{w}^* \right\|^2 \right] + \eta_t^2 G^2.$$

where $\Pi_{\mathcal{W}}()$ means the projection, and the second inequality holds due to L-smoothness. Recall that for a one-shot federated learning setting, we have $\mathbf{w}_0^{k'} = \mathbf{w}_0$. Then, for any $mE \leq t < (m+1)E$, unrolling the recursion, we have,

$$\mathbb{E}\left[ \left\| \mathbf{w}_t^{k'} - \mathbf{w}^* \right\|^2 \right] \leq \prod_{t=0}^{E-1} (1 - 2\eta_t L)^t \mathbb{E}\left[ \left\| \mathbf{w}_0^{k'} - \mathbf{w}^* \right\|^2 \right] + G^2 \sum_{t=0}^{E-1} (1 - 2\eta_t L)^t \eta_t^2$$

$$\leq \prod_{t=0}^{E-1} (1 - 2\eta_t L)^t \frac{4G^2}{\mu^2} + G^2 \sum_{t=0}^{E-1} (1 - 2\eta_t L)^t \eta_t^2.$$

The last inequality holds because of Lemma A.3.

**Lemma A.3.** *(Lemma 2 of Rakhlin (Rakhlin et al., 2011)). If Assumption 3.1 and Assumption 3.2 hold, then*

$$\mathbb{E}[||\mathbf{w}_0 - \mathbf{w}^*||^2] \leq \frac{4G^2}{\mu^2}.$$

□

A.6   More discussions about examples

This subsection provides two additional examples demonstrating the broad applicability of our Lemma 4.2 to existing convergence results. To ensure better understanding, we adjust these examples' prerequisites and parameter definitions to fit in our setting, and lay out the key convergence findings from these studies (Yang et al., 2021; Karimireddy et al., 2020). For more details, interested readers can refer to the corresponding references.

**Example 2.** *(Lemma 7 in (Karimireddy et al., 2020)). Suppose the functions satisfy Assumptions 3.1-3.4 and $\frac{1}{N}\sum_{i=1}^{N}\|\nabla f_i(\mathbf{w})\|^2 \leq G^2 + B^2\|\nabla f(\mathbf{w})\|^2$ (Bounding gradient similarity). For any step-size (local(global) learning rate $\eta_l(\eta_g)$) satisfying $\eta_l \leq \frac{1}{(1+B^2)8LE\eta_g}$ and effective step-size $\tilde{\eta} \triangleq E\eta_g\eta_l$, the updates of FedAvg satisfy*

$$
\begin{aligned}
\mathbb{E}\left\|\mathbf{w}^t - \mathbf{w}^\star\right\|^2 \leq{} & \left(1 - \frac{\mu\tilde{\eta}}{2}\right)\mathbb{E}\left\|\mathbf{w}^{t-1} - \mathbf{w}^\star\right\|^2 + \left(\frac{1}{ES}\right)\tilde{\eta}^2\sigma^2 \\
& + \left(1 - \frac{S}{N}\right)4\tilde{\eta}^2 G^2 - \tilde{\eta}\left(\mathbb{E}\left[f\left(\mathbf{w}^{t-1}\right)\right] - f\left(\mathbf{w}^\star\right)\right) + 3L\tilde{\eta}\mathcal{E}_t
\end{aligned}
\tag{6}
$$

*where $S$ ($N$) indicates the number of sampled (total) agents, $E$ is the synchronization interval (local steps) and $\mathcal{E}_t$ is the drift caused by the local updates on the clients defined to be*

$$
\mathcal{E}_t \triangleq \underbrace{\frac{1}{EN}\sum_{e=1}^{E}\sum_{i=1}^{N}\mathbb{E}_t\left[\left\|\mathbf{w}_{i,k}^t - \mathbf{w}^{t-1}\right\|^2\right]}_{\text{Divergence term}}
$$

*Clearly, we can readily replace the original term $\mathcal{E}_t$ with Lemma 4.2, and subsequently rewrite the convergence result (Theorem 1 of Karimireddy (Karimireddy et al., 2020)) to introduce the degree of non-iid.*

Prior to presenting Example 3, we initially introduce an extra assumption they have made, which is analogous to Assumption A.2.

**Assumption A.1.** *(Assumption 3 in (Yang et al., 2021)) (Bounded Local and Global Variances) The variance of each local stochastic gradient estimator is bounded by $\mathbb{E}\left[\left\|\nabla f_i\left(\mathbf{w}, \xi^i\right) - \nabla f_i(\mathbf{w})\right\|^2\right] \leq \sigma_L^2$, and the global variability of local gradient is bounded by $\mathbb{E}\left[\|\nabla f_i(\mathbf{w}) - \nabla f(\mathbf{w})\|^2\right] \leq \sigma_G^2, \forall i \in [N]$.*

**Example 3.** *(Theorem 1 in (Yang et al., 2021)). Let local and global learning rates $\eta_L$ and $\eta$ satisfy $\eta_L \leq \frac{1}{8LE}$ and $\eta\eta_L \leq \frac{1}{EL}$. Under Assumptions 3.2, 3.4, A.1, and with full agent participation, the sequence of outputs $\{\mathbf{w}_k\}$ generated by its algorithm satisfies:*

$$
\min_{t\in[T]}\mathbb{E}\left[\|\nabla f\left(\mathbf{w}_t\right)\|_2^2\right] \leq \frac{f\left(\mathbf{w}_0\right) - f\left(\mathbf{w}^*\right)}{c\eta\eta_L ET} + \frac{1}{c}\left[\frac{L\eta\eta_L}{2N}\sigma_L^2 + \frac{5E\eta_L^2 L^2}{2}\left(\sigma_L^2 + 6E\sigma_G^2\right)\right]
\tag{7}
$$

*where $E$ is the synchronization interval (local steps) and $c$ is a constant. Here, we can define the constant $\sigma_G^2 \triangleq 4G^2\delta_k^2$ using Eq.(8) to introduce the non-iid degree $\delta_k^2$. Note that the above result is obtained within a non-convex setting. One can change the right-hand of Eq. (7) into $\mathbb{E}\left[f\left(\bar{\mathbf{w}}_T\right) - f\left(\mathbf{w}^*\right)\right]$ by using strong convexity condition $\|\nabla f(\mathbf{w})\|_2^2 \geq 2\mu\left(f(\mathbf{w}) - f^*\right)$.*

A.7   Discussions about bounding global gradient variance

Note that there exists a common assumption in (Yang et al., 2021; 2022; Wang et al., 2019; Stich, 2018) about bounding gradient dissimilarity or variance using constants. To illustrate the broad applicability of the proposed non-iid degree metric, we further discuss how to rewrite this type of assumption by leveraging $\delta_k$, that is, re-measuring the gradient discrepancy among agents, *i.e.*, $\|\nabla F_k(\mathbf{w}) - \nabla F(\mathbf{w})\|^2, \forall k \in [N]$. As a reminder, we present it in Assumption A.2.

**Assumption A.2.** *(Bounded Global Variance) The global variability of the local gradient of the loss function is bounded by $\mathbb{E}[\|\nabla F_k(\mathbf{w}) - \nabla F(\mathbf{w})\|^2] \leq \sigma^2, \forall k \in [N]$.*

Below, we will show that this quantity also has a close connection with the Wasserstein distance $\delta_k$, that is,

$$
\|\nabla F_k(\mathbf{w}) - \nabla F(\mathbf{w})\|^2 \leq 4G^2\delta_k^2, \quad \forall k \in [N].
\tag{8}
$$

Note that

$$
\begin{aligned}
\|\nabla F_k(\mathbf{w}) - \nabla F(\mathbf{w})\| &= \left\| \sum_{\zeta \in \mathcal{D}} \nabla_{\mathbf{w}} l(\mathbf{w}, \zeta) \left( p^{(k)}(\zeta) - p^{(c)}(\zeta) \right) \right\| \\
&\leq \sum_{\zeta \in \mathcal{D}} \|\nabla_{\mathbf{w}} l(\mathbf{w}, \zeta)\| \left| p^{(k)}(\zeta) - p^{(c)}(\zeta) \right| \\
&\leq G \sum_{i=1}^{I} \left| p^{(k)}(y = i) - p^{(c)}(y = i) \right| = 2G\delta_k,
\end{aligned}
$$

where the first inequality holds due to Jensen inequality, and the second inequality holds because of the assumption that $\|\nabla_{\mathbf{w}} l(\mathbf{w}, z)\| \leq G_k \leq G$ for any $i$ and $w$, which analogous to Assumption 3.4. We also discretely split the training data according to their labels $i, \forall i \in [I]$. Here, we make a mild assumption that the aggregated gradients $\nabla F(\mathbf{w})$ of one communication round can be approximately regarded as the gradients resulting from a reference distribution $p^{(c)}$. Therefore, we can use the distribution $p^{(c)}$ as the corresponding distribution of $\nabla F(\mathbf{w})$.

# B    OMITTED PROOFS IN SECTION 5

We first provide more discussion about Assumption 5.1 and present omitted proofs of theorems and lemmas in Section 5.

## B.1    MORE DISCUSSIONS ABOUT ASSUMPTION 5.1

Here, we first discuss the core requirement of the payment function. It is easily to think that this assumption holds if $\frac{\partial f}{\partial e_k} > 0$ and $\frac{\partial^2 f}{\partial e_k^2} < 0$. The first-order partial derivative of the payment function is

$$
\frac{\partial f}{\partial e_k} = f'\left( \frac{Q}{\Phi\delta_k^2 + \Phi\delta_{k'}^2 + \Upsilon} \right) \cdot \frac{-2\Phi Q \delta_k \delta_k'}{\left( \Phi\delta_k^2 + \Phi\delta_{k'}^2 + \Upsilon \right)^2}.
$$

For simplicity, we replace $\delta_k(e_k)$ with $\delta_k$. Similarly, the second-order partial derivative of the payment function is,

$$
\begin{aligned}
\frac{\partial^2 f}{\partial e_k^2} =& f''\left( \frac{Q}{\Phi\delta_k^2 + \Phi\delta_{k'}^2 + \Upsilon} \right) \cdot \frac{4\Phi^2 Q^2 \delta_k^2 (\delta_k')^2}{\left( \Phi\delta_k^2 + \Phi\delta_{k'}^2 + \Upsilon \right)^4} \\
&+ f'\left( \frac{Q}{\Phi\delta_k^2 + \Phi\delta_{k'}^2 + \Upsilon} \right) \cdot \frac{-2\Phi Q((\delta_k')^2 + \delta_k \delta_k'')(\Phi\delta_k^2 + \Phi\delta_{k'}^2 + \Upsilon) + 4\Phi^2 Q \delta_k^2 (\delta_k')^2}{\left( \Phi\delta_k^2 + \Phi\delta_{k'}^2 + \Upsilon \right)^3} \\
=& f''\left( \frac{Q}{\Phi\delta_k^2 + \Phi\delta_{k'}^2 + \Upsilon} \right) \cdot \frac{4\Phi^2 Q^2 \delta_k^2 (\delta_k')^2}{\left( \Phi\delta_k^2 + \Phi\delta_{k'}^2 + \Upsilon \right)^4} \\
&+ f'\left( \frac{Q}{\Phi\delta_k^2 + \Phi\delta_{k'}^2 + \Upsilon} \right) \cdot \frac{-2Q\Phi[(\Phi\delta_{k'}^2 + \Upsilon - \Phi\delta_k^2)(\delta_k')^2 + (\Phi\delta_{k'}^2 + \Upsilon + \Phi\delta_k^2)\delta_k \delta_k'']}{\left( \Phi\delta_k^2 + \Phi\delta_{k'}^2 + \Upsilon \right)^3}.
\end{aligned}
$$

Here, we discuss two payment function settings in terms of linear function and logarithmic function. In particular, note that parameters $\Phi$ and $\Upsilon$ are both greater than 0, that is, $\Phi > 0$ and $\Upsilon > 0$. More discussions about these two parameters are presented in Section 6.

**Linear function.** If the payment function is a linear non-decreasing function, then $f'(\cdot)$ is a constant $M > 0$, and $f''(\cdot) = 0$. Thus, we can simplify the above constraints as follows.

$$
\frac{\partial f}{\partial e_k} = M \cdot \frac{-2\Phi Q \delta_k \delta_k'}{\left( \Phi\delta_k^2 + \Phi\delta_{k'}^2 + \Upsilon \right)^2} > 0,
$$

$$
\frac{\partial^2 f}{\partial e_k^2} = 0 + M \cdot \frac{-2Q\Phi[(\Phi\delta_{k'}^2 + \Upsilon - \Phi\delta_k^2)(\delta_k')^2 + (\Phi\delta_{k'}^2 + \Upsilon + \Phi\delta_k^2)\delta_k \delta_k'']}{\left( \Phi\delta_k^2 + \Phi\delta_{k'}^2 + \Upsilon \right)^3} < 0.
$$

Note that $\delta_k' \triangleq \frac{d\delta_k(e_k)}{de_k} < 0$ always holds due to the fact that more effort, less data heterogeneous. With this in mind, there is no need to ask for any extra requirement to satisfy $\frac{\partial f}{\partial e_k} > 0$. Therefore, for a linear non-decreasing payment function, if the inequality $(\Phi\delta_{k'}^2 + \Upsilon - \Phi\delta_k^2)(\delta_k')^2 + (\Phi\delta_{k'}^2 + \Upsilon + \Phi\delta_k^2)\delta_k \delta_k'' > 0$ holds, then Assumption 5.1 holds.

**Logarithmic function.** Now we consider the case with a logarithmic function. The corresponding inequality can be expressed as follows.

$$\frac{\partial f}{\partial e_k} = \frac{\Phi\delta_k^2 + \Phi\delta_{k'}^2 + \Upsilon}{Q} \cdot \frac{-2\Phi Q\delta_k\delta_k'}{\left(\Phi\delta_k^2 + \Phi\delta_{k'}^2 + \Upsilon\right)^2} = \frac{-2\Phi\delta_k\delta_k'}{\Phi\delta_k^2 + \Phi\delta_{k'}^2 + \Upsilon} > 0,$$

$$\frac{\partial^2 f}{\partial e_k^2} = \frac{\left(\Phi\delta_k^2 + \Phi\delta_{k'}^2 + \Upsilon\right)^2}{Q^2} \cdot \frac{4\Phi^2 Q^2 \delta_k^2 (\delta_k')^2}{\left(\Phi\delta_k^2 + \Phi\delta_{k'}^2 + \Upsilon\right)^4}$$

$$+ \frac{\Phi\delta_k^2 + \Phi\delta_{k'}^2 + \Upsilon}{Q} \cdot \frac{-2Q\Phi[(\Phi\delta_{k'}^2 + \Upsilon - \Phi\delta_k^2)(\delta_k')^2 + (\Phi\delta_{k'}^2 + \Upsilon + \Phi\delta_k^2)\delta_k\delta_k'']}{\left(\Phi\delta_k^2 + \Phi\delta_{k'}^2 + \Upsilon\right)^3}$$

$$= \frac{4\Phi^2 \delta_k^2 (\delta_k')^2}{\left(\Phi\delta_k^2 + \Phi\delta_{k'}^2 + \Upsilon\right)^2} + \frac{-2\Phi[(\Phi\delta_{k'}^2 + \Upsilon - \Phi\delta_k^2)(\delta_k')^2 + (\Phi\delta_{k'}^2 + \Upsilon + \Phi\delta_k^2)\delta_k\delta_k'']}{\left(\Phi\delta_k^2 + \Phi\delta_{k'}^2 + \Upsilon\right)^2}$$

$$= 2\Phi \cdot \frac{(3\Phi\delta_k^2 - \Phi\delta_{k'}^2 - \Upsilon)(\delta_k')^2 - (\Phi\delta_{k'}^2 + \Upsilon + \Phi\delta_k^2)\delta_k\delta_k''}{\left(\Phi\delta_k^2 + \Phi\delta_{k'}^2 + \Upsilon\right)^2} < 0.$$

Similarly, Assumption 5.1 holds if $(3\Phi\delta_k^2 - \Phi\delta_{k'}^2 - \Upsilon)(\delta_k')^2 - (\Phi\delta_{k'}^2 + \Upsilon + \Phi\delta_k^2)\delta_k\delta_k'' < 0$.

## B.2 PROOF OF THEOREM 5.1

**Theorem 5.1.** (Optimal effort level). Consider agent $k$ with its marginal cost and the payment function inversely proportional to the generalization error gap with any randomly selected peer $k'$. Then, agent $k$'s optimal effort level $e_k^*$ is:

$$e_k^* = \begin{cases} 0, & \text{if} \quad \max_{e_k \in [0,1]} u_k(e_k, e_{k'}) \le 0; \\ \hat{e}_k \quad \text{such that} \quad \partial f(\hat{e}_k, e_{k'})/\partial\delta_k(\hat{e}_k) + c_k d'(\delta_k(\hat{e}_k)) = 0, & \text{otherwise.} \end{cases}$$

*Proof.* Given the definition of utility functions, we have:

$$\partial u_k(e_k)/\partial e_k = \partial f(e_k, e_{k'})/\partial e_k + c_k d'(\delta_k(\hat{e}_k)). \tag{9}$$

It is easy to see that if $u_k(e_k, e_{k'}) \le 0$, then agent $k$ will have no motivation to invest efforts, *i.e.*, $e_k^* = 0$. In this regard, it can be categorized into two cases according to the value of the cost coefficient $c_k$.

**Case 1 (high-cost agent):** If the marginal cost $c_k$ is too high, the utility of agents could still be less than 0, that is, $\max_{e_k \in [0,1]} u_k(e_k, e_{k'}) \le 0$. In this case, agent $k$ will make no effort, *i.e.*, $e_k^* = 0$.

**Case 2 (low-cost agent):** Otherwise, there exists a set of effort levels $\hat{e}_k$, which can achieve non-negative utility.

If Assumption 5.1 holds and there exists a set of effort levels $\hat{e}_k$ with non-negative utility, we can easily derive the optimal effort level by using Eq. (9). And the optimal effort level that maximizes the utility can be obtained when $\partial f(\hat{e}_k, e_{k'})/\partial\hat{e}_k + c_k d'(\delta_k(\hat{e}_k)) = 0$.

Therefore, we finished the proof of Theorem 5.1. $\qquad\square$

## B.3 PROOF OF THEOREM 5.2

Before diving into the proof of Theorem 5.2, we first address the question of whether the utility functions exhibit well-behaved characteristics, and if not, what conditions must be met to ensure such a property.

**Lemma B.1.** *If Assumption 5.1 holds, the utility function $u_k(\cdot)$ is a well-behaved function, which is continuously differentiable and strictly-concave on $\boldsymbol{e}$.*

*Proof.* Here, we certify that $u_k(e_k)$ is a well-behaved function. According to Assumption 5.1, we can get the maximum and minimum value of $\partial f(e_k, e_{k'})/\partial e_k$ when $e_k = 0$ and $e_k = 1$, respectively. Note that $\partial f(e_k, e_{k'})/\partial e_{k'}$ is the same as $\partial f(e_k, e_{k'})/\partial e_k$. Therefore,

$$\partial u_k(\boldsymbol{e})/\partial e_k = \partial f(e_k, e_{k'})/\partial e_k + c_k d'(\delta_k) \ge \left[\partial f(e_k, e_{k'})/\partial e_k + c_k d'(\delta_k)\right]_{e_k=1} \triangleq d_k^2$$

$$\partial u_k(\boldsymbol{e})/\partial e_{k'} = \partial f(e_k, e_{k'})/\partial e_{k'} \le \left[\partial f(e_k, e_{k'})/\partial e_k + c_k d'(\delta_k)\right]_{e_k=0} \triangleq d_{k'}^1$$

Note that the derivative of the utility function $u_k(\boldsymbol{e})$ with respect to $e_k$ has an identical form to the derivative of $u_k(\boldsymbol{e})$ with respect to $e_{k'}$. Given this, and in light of the discussion presented in Appendix B.1, it becomes

clear that the utility function $u_k(\cdot)$ is continuously differentiable with respect to $\boldsymbol{e}$. Integrating Lemma B.1 and the discussion about Assumption 5.1, it becomes clear that the utility function $u_k(\cdot)$ is not only continuously differentiable but also strictly concave with respect to $\boldsymbol{e}$. Thus, we can finish this part of the proof.  □

We emphasize that it is evident that our utility functions are naturally well-behaved, supporting the existence of equilibrium. To achieve this, we resort to Brouwer's fixed-point theorem (Brouwer, 1911), which yields the existence of the required fixed point of the best response function.

**Theorem 5.2.** (Existence of pure Nash equilibrium). Denote by $\boldsymbol{e}^*$ the optimal effort level, if the utility functions $u_k(\cdot)$s are well-behaved over the set $\prod_{k\in[N]}[0,1]$, then there exists a pure Nash equilibrium in effort level $\boldsymbol{e}^*$ which for any agent $k$ satisfies,

$$u_k(e_k^*, \boldsymbol{e}_{-k}^*) \geq u_k(e_k, \boldsymbol{e}_{-k}^*), \quad \text{for all } e_k \in [0,1], \forall k \in [N].$$

*Proof.* For a set of effort levels $\boldsymbol{e}$, define the best response function $\boldsymbol{R}(\boldsymbol{e}) \triangleq \{R_k(\boldsymbol{e})\}_{\forall k \in [N]}$, where

$$R_k(\boldsymbol{e}) \triangleq \arg \max_{\tilde{e}_k \in [0,1]} \{u_k(\tilde{e}_k, \boldsymbol{e}_{-k}) \triangleq f(\tilde{e}_k, \boldsymbol{e}_{-k}) - c_k |\delta_k(0) - \delta_k(e_k)| \geq 0\}, \tag{10}$$

where recall that $f(\tilde{e}_k, \boldsymbol{e}_{-k})$ is the payment function inversely proportional to the performance gap between agent $k$ and a randomly selected agent (peer) $k'$ with their invested data distributions $\mathcal{D}_k(e_k), \mathcal{D}_k'(e_{k'})$, respectively. Notice that, by definition, the payment function is merely relevant to a randomly selected agent (peer) $k'$, not for all agents. Here, without loss of generality, we extend the range from a single agent to all agents, that is, rewrite the payment function in the form of $f(e_k, \boldsymbol{e}_{-k})$.

If there exists a fixed point to the mapping function $R(\cdot)$, *i.e.*, there existed $\tilde{\boldsymbol{e}}$ such that $\tilde{\boldsymbol{e}} \in R(\tilde{\boldsymbol{e}})$. Then, we can say that $\tilde{\boldsymbol{e}}$ is the required equilibrium effort level by the definition of the mapping function. Therefore, the only thing we need to do is to prove the existence of the required fixed points. We defer this part of the proof in Lemma B.2 by applying Brouwer's fixed-point theorem.

**Lemma B.2.** *The best response function $R(\cdot)$ has a fixed point,* i.e., *$\exists \boldsymbol{e}^* \in \prod_{k=1}^N [0,1]$, such that $\boldsymbol{e}^* = R(\boldsymbol{e}^*)$, if the utility functions over agents are well-behaved.*

*Proof.* First, we introduce the well-known Brouwer's fixed point theorem.

**Lemma B.3.** *(Brouwer's fixed-point theorem (Brouwer, 1911)) Any continuous function on a compact and convex subset of $R : \prod_{k\in[N]}[0,1] \to \prod_{k\in[N]}[0,1]$ has a fixed point.*

In our case, Lemma B.3 requires that the best response function $R$ on a compact and convex subset be continuous. Notice that the finite product of compact, convex, and non-empty sets $\prod_{k\in[N]}[0,1]$ is also compact, convex, and non-empty. Therefore, $R$ is a well-defined map from $\prod_{k\in[N]}[0,1]$ to $\prod_{k\in[N]}[0,1]$, which is a subset of $\mathbb{R}^N$ that is also convex and compact. Therefore, there is only one thing left to prove: $R$ is a continuous function over $\prod_{k\in[N]}[0,1]$.

More formally, we show that for any $\boldsymbol{\tau} \in \mathbb{R}^N$ with $\|\boldsymbol{\tau}\|_1 \leq 1$, $\lim_{\varepsilon \to 0} |R_k(\boldsymbol{e} + \varepsilon\boldsymbol{\tau}) - R_k(\boldsymbol{e})| = 0$. For simplicity, we define $\boldsymbol{e}' = \boldsymbol{e} + \varepsilon\boldsymbol{\tau}$, $x = R_k(\boldsymbol{e})$, and $x' = R_k(\boldsymbol{e}')$. Integrating Lemma B.1 and the discussion about Assumption 5.1, it becomes clear that the utility function $u_k(\cdot)$ is not only continuously differentiable but also strictly concave with respect to $\boldsymbol{e}$. Therefore, there exists a unique solution $x = R_k(\boldsymbol{e})$ to the first-order condition $\partial u_k(e_k, \boldsymbol{e}_{-k})/\partial e_k = 0$ for each $\boldsymbol{e}$. By the Implicit Function theorem, if $\partial^2 u_k(e_k, \boldsymbol{e}_{-k})/\partial e_k^2 \neq 0$, then the solution to the equation $\partial u_k(e_k, \boldsymbol{e}_{-k})/\partial e_k = 0$ is a function of $\boldsymbol{e}$ that is continuous. Therefore, it is obvious that $\lim_{\varepsilon \to 0} |R_k(\boldsymbol{e} + \varepsilon\boldsymbol{\tau}) - R_k(\boldsymbol{e})| = 0$ if $|\boldsymbol{e}' - \boldsymbol{e}| \leq \varepsilon\|\boldsymbol{\tau}\|_1$ and $\|\boldsymbol{\tau}\|_1 \leq 1$. Therefore, $R$ is continuous over the set $\prod_{k\in[N]}[0,1]$. Overall, the remaining proof follows according to Brouwer's fixed point theorem.  □

Therefore, as we previously stated, this fixed point $\boldsymbol{e}^*$ of the best response function $R(\cdot)$ also serves as the equilibria of our framework.  □

**Remark 1.** *(Existence of other possible equilibriums) Note that no one participating is a possible equilibrium in our setting when the marginal cost is too high. In this case, all agents' optimal effort levels will be 0. Another typical equilibrium in federated learning, known as free-riding, does not exist in our setting.*

*Proof.* Here, we discuss two typical equilibriums in terms of no one participating and free-riding. Note that no one participating is a possible equilibrium in our setting when the marginal cost is too high. In this case, all agents' optimal effort levels will be 0. Then, we further provide a brief discussion of the free-riding problem. Given Theorem 5.2, there is always a Nash equilibrium in effort level that no agent can improve their utility

by unilaterally changing their contribution. If all players are rational (and such an equilibrium is unique), then such a point is a natural attractor with all the agents gravitating towards such contributions. Therefore, we can use this property to prove that free-riding will not happen in our setting. To avoid free-riding, a reasonable goal for a mechanism designer is to maximize the payoff of the learner when all players are contributing such equilibrium amounts (Karimireddy et al., 2022). Employing a two-stage Stackelberg game is inherently apt for circumventing the free-riding problem in our paper. Unlike the traditional free-riding scenario where all agents are rewarded directly based on the model's accuracy (represented by the convergence bound in our case, which is the underlying cause of free-riding), our game implements a scoring function. This function rewards agents by using a coefficient $Q$ that depends on the convergence bound. Consequently, the newly introduced coefficient $Q$ decouples the direct connection between the model's performance and the agents' rewards. The learner's optimal response is then determined by solving the Problem 2. This ensures the goal of maximizing the learner's payoff and the uniqueness of the equilibrium, thereby preventing free-riding. □

**Remark 2.** *(The consistency of extending to all agents) Note that the introduced randomness can serve as an effective tool to circumvent the coordinated strategic behaviors of most agents. Then, the consistency of extending to all agents is still satisfied.*

*Proof.* Note that inspired by peer prediction, randomness serves as an effective tool to circumvent the coordinated strategic behaviors of most agents. There is a consistency between randomly selecting one peer agent and the extension that uses the mean of the models. We can verify this consistency by using the mean of models $\overline{\mathbf{w}}$ returned by all agents to substituting $\mathbf{w}$. Here, we present a high-level sketch of our idea, utilizing inequalities similar to those found in Appendix A.3 (Proof of Theorem 4.3). For any confusion, please refer to Appendix A.3.

$$F_c(\mathbf{w}^k) - F_c(\overline{\mathbf{w}}) \leq \langle \nabla F_c(\overline{\mathbf{w}}), \mathbf{w}^k - \overline{\mathbf{w}} \rangle + \frac{L}{2}\|\mathbf{w}^k - \overline{\mathbf{w}}\|^2$$

$$\leq L\|\mathbf{w}^k - \overline{\mathbf{w}}\|^2 + \frac{L}{2}\|\overline{\mathbf{w}} - \mathbf{w}^*\|^2$$

$$\leq \frac{L}{N}\sum_{k'=1}^{N}\left(\|\mathbf{w}^k - \mathbf{w}^{k'}\|^2 + \frac{1}{2}\|\mathbf{w}^{k'} - \mathbf{w}^*\|^2\right).$$

Then, by integrating the insights from Lemma A.1 and Lemma A.2, we can get a similar result of Theorem 4.3. Therefore, the consistency holds. □

# C EXPERIMENT DETAILS AND ADDITIONAL RESULTS

## C.1 PARAMETER SETTINGS IN EVALUATION

**Data split.** Instead of splitting the entire dataset into several local ones, we're employing a method similar to (McMahan et al., 2017). From the whole training dataset, we will randomly pick a set number of data samples, specifically 500/600 (according to different datasets), for every round that agents used to train models. In particular, to introduce statistical heterogeneity, we partitioned the data based on the image classes each agent owns. Suppose that there are $N = 10$ agents who fully participate in the training process. Therefore, for each round, the agent will continuously re-sample the data points with the fixed sample size, such that each agent contains the same quantity of data samples but with varying proportions of different image classes. Specifically, for each agent, a proportion of $\delta$ data samples is from a majority class, while the remaining samples are uniformly distributed across the rest of the classes. For instance, for each agent, $\delta = 0.8$ means that $80\%$ of the data samples are from one class (label), and the remaining $20\%$ belong to the other nine classes (labels). In this way, we can leverage this metric $\delta$ to measure the degree of non-iid qualitatively, while also capturing the idea of Wasserstein distance.

We apply optimizer Adam (Kingma & Ba, 2015) with 0.9 momentum rate to optimize the model, and use BatchNormalization (Ioffe & Szegedy, 2015) for regularization. We specify parameter settings of three datasets, and model architectures in Tables 1-3.

## C.2 PERFORMANCE RESULTS USING OTHER NON-IID METRICS

Many works prefer using the number of classes as a non-iid metric for data partitioning (Yang et al., 2022), that is, using the number of classes $p$ within each client's local dataset, even though this metric may lack flexibility in adjusting the degree of non-iid. Here, to illustrate the heterogeneous efforts, we also utilize the number of classes (typical) as a heterogeneity metric, despite its rigidity in altering non-iid degrees. The corresponding results are shown in Figure 5 and Figure 6, which align with the previous findings in Section 6.2.

Table 1: CNN Architecture for MNIST.

| Layer Type | Size |
|---|---|
| Convolution + ReLu | $1 \times 20 \times 5$ |
| Max Pooling | $2 \times 2$ |
| Convolution + ReLu | $20 \times 50 \times 5$ |
| Max Pooling | $2 \times 2$ |
| Fully Connected + ReLU | $(4 \times 4 \times 50) \times 500$ |
| Fully Connected | $500 \times 10$ |

Table 2: CNN Architecture for FashionMNIST.

| Layer Type | Size |
|---|---|
| Convolution + BatchNorm + ReLu | $1 \times 16 \times 5$ |
| Max Pooling | $2 \times 2$ |
| Convolution + BatchNorm + ReLu | $16 \times 32 \times 5$ |
| Max Pooling | $2 \times 2$ |
| Fully Connected | $(7 \times 7 \times 32) \times 10$ |

Table 3: CNN Architecture for CIFAR-10 and CIFAR-100.

| Layer Type | Size |
|---|---|
| Convolution + ReLu | $3 \times 6 \times 5$ |
| Max Pooling | $2 \times 2$ |
| Convolution + ReLu | $6 \times 6 \times 16$ |
| Max Pooling | $2 \times 2$ |
| Fully Connected + ReLU | $(16 \times 5 \times 5) \times 120$ |
| Fully Connected + ReLU | $120 \times 84$ |
| Fully Connected | $84 \times 10(20)$ |

**Data Partitioning.** Then, we also evaluate a common data partitioning method, that is, evenly partitioning the whole dataset for agents, and then sampling from local datasets. More specifically, we will employ both uniform and long-tail distributions to determine the data proportions of minority classes, excluding the majority class, to assess their varying impacts. In the uniform distribution setting, Figure. 7 and 8 present the corresponding results, which are consistent with previous results. However, these heterogeneous efforts may be not so obvious. This is because we typically use a part of the local data for training in each round, and this may unintentionally even out the heterogeneity.

Besides, Figure. 9 and 10 present the performance results in the long-tail distribution setting. We can observe that the results shown in Figure. 9 and 10 are consistent with previously results.

## C.3 SCORING FUNCTION SETTINGS

For linear functions, we have

$$f(e_k, e_{k'}) = \kappa \frac{Q}{\Phi \delta_k^2(e_k) + \Phi \delta_{k'}^2(e_{k'}) + \Upsilon}$$

According to $\partial u_k(e_k)/\partial \delta_k(e_k) = \partial f(e_k, e_{k'})/\partial \delta_k(e_k) + c_k = 0$, we can derive the optimal effort level by addressing the following equation,

$$\delta_k^4 + 2(\delta_{k'}^2 + \frac{\Upsilon}{\Phi})\delta_k^2 - 2\frac{\kappa Q}{c_k \Phi}\delta_k + \frac{(\Phi \delta_{k'}^2 + \Upsilon)^2}{\Phi} = 0$$

For brevity, we use $\delta_k$ to represent $\delta_k(e_k)$ as a decision variable. Here, $\delta_{k'}$ can be regarded as a constant. Then, for logarithmic functions, we define it as

$$f(e_k, e_{k'}) = \log(\frac{Q}{\Phi \delta_k^2(e_k) + \Phi \delta_{k'}^2(e_{k'}) + \Upsilon})$$

Similarly, we can easily derive and obtain the optimal effort level by addressing Eq. (11). For simplicity, we take a linear function to simulate $d(\cdot)$ shown in the form of the cost function. Here, we generally rewrite it into

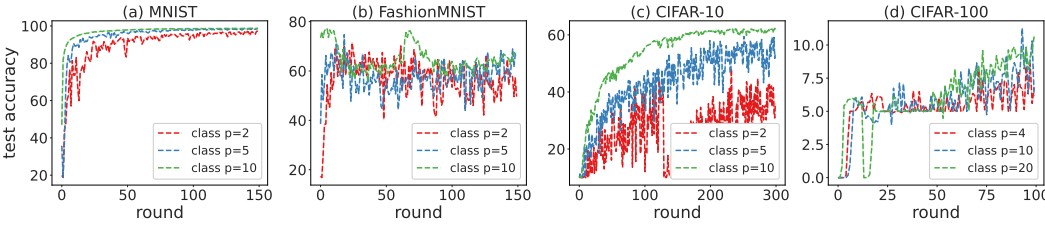

Figure 5: (**The number of classes**) FL training process under different non-iid degrees.

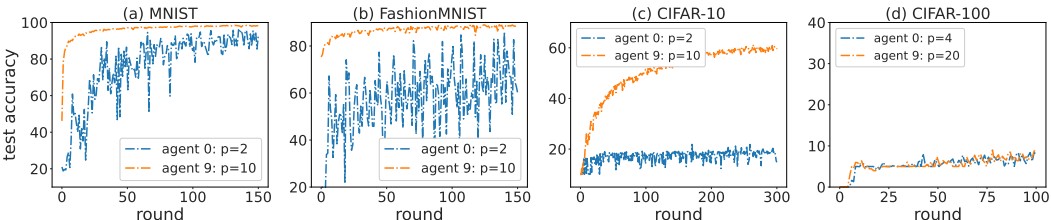

Figure 6: (**The number of classes**) Performance comparison with peers. In default, the non-iid degree is 0.5.

a general form $\delta_k \triangleq f(\delta_{k'}^2), \forall k$, to calculate the corresponding $\delta_k$ for agents.

$$\delta_k^2 - \frac{2}{c_k}\delta_k + \delta_{k'}^2 + \frac{\Upsilon}{\Phi} = 0 \implies \delta_k \triangleq f(\delta_{k'}^2) = \frac{1}{c_k} \pm \sqrt{\frac{1}{c_k^2} - \delta_{k'}^2 - \frac{\Upsilon}{\Phi}} \tag{11}$$

where $\delta_{k'}$ indicates the non-iid degree of a randomly selected peer. Note that there are two results returned by $f(\delta_{k'}^2)$. Basically, we will choose the smaller one (smaller effort level).

**Lipschitz constant.** Here, we focus more on the image classification problem with classical cross-entropy loss. In general, the Lipschitz constant of the cross-entropy loss function is determined by the maximum and minimum probabilities of the output of the network. Specifically, for a neural network with the output probability distribution $\hat{y}$ and true distribution $y$, the Lipschitz constant of the cross-entropy loss function with respect to the L1-norm can be lower-bounded as

$$L \geq \sup\{\|\frac{\partial CE(\hat{y}_n, y_n)}{\partial \hat{y}_n}\|, \forall x_n\} \implies L \geq \max_{i \in I}\{|\frac{y_i}{\hat{y}_i}|\}$$

where $\{(x_n, y_n)\}_{n=1}^N$ indicates $N$ samples, and $i \in I$ denotes the index of possible labels. We assume that the corresponding averaged error rate of other labels is 0.01 approximately. Consider the worse case where the ML model misclassifies one sample, which means the probability that the predicted label is the true label is equal to the average error rate. Then, we can derive the bound of the Lipschitz constant using $L \geq \max_{i \in I}\{|\frac{p_i}{q_i}|\} = \frac{1}{0.01} = 100$. Here, we set the default value of $L$ as 100. Recall that the learning rate $\eta = 0.01$, and assume $\eta_t = \eta, \forall t$. Therefore, we can simplify parameter $\Phi$ as

$$\Phi = 2L^2 G^2 \sum_{t=0}^{E-1} (\eta_t^2(1 + 2\eta_t^2 L^2))^t \leq 2L^2 G^2 \sum_{t=0}^{E-1} 3\eta^2 = 6EG^2.$$

Similarly, for parameter $\Upsilon$, we have

$$\Upsilon = \prod_{t=0}^{E-1}(1 - 2\eta L)^t \frac{2G^2 L}{\mu^2} + \frac{LG^2}{2} \sum_{t=0}^{E-1}(1 - 2\eta L)^t \eta^2 = \prod_{t=0}^{E-1}(-1)^t \frac{2G^2 L}{\mu^2} + \frac{L\eta^2 G^2}{2} \sum_{t=0}^{E-1}(-1)^t$$

Here, we have

$$\Upsilon = \begin{cases} \frac{2G^2}{\mu}, & \text{if } E \text{ is even;} \\ -\frac{2G^2}{\mu} - \frac{\eta G^2}{2}, & o.w. \end{cases}$$

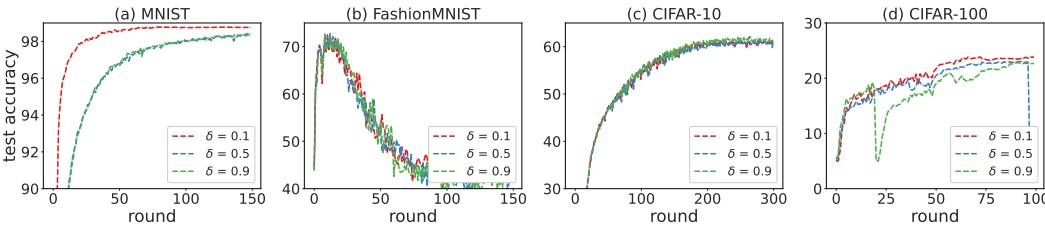

Figure 7: (**Uniform distribution setting**) FL training process under different non-iid degrees.

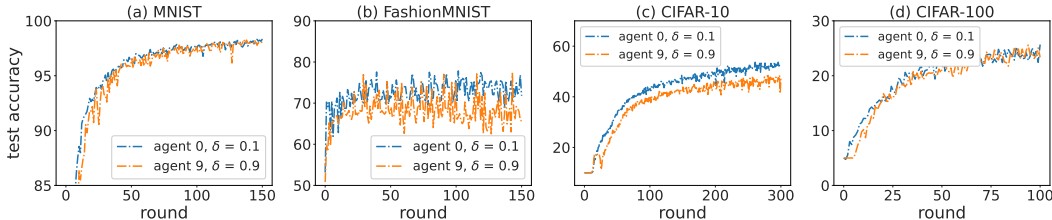

Figure 8: (**Uniform distribution setting**) Performance comparison with peers. In default, the non-iid degree is 0.5.

In general, we discuss a specific case that $E$ is even, *i.e.*, $\Upsilon = \frac{2G^2}{\mu}$ to guarantee $\Upsilon$ is non-negative. Here, suppose that the convex constant $\mu = 0.01$. Recall that $G$ is the upper bound of the gradient on a random sample shown in Assumption 3.4.

## C.4 ADDITIONAL RESULTS

**Gradient divergence.** To echo the effectiveness of bounding global gradient variance, we evaluate the global gradient variance. Here, we present the norm of gradients of two randomly selected agents. Figure 11 illustrates that the gradient gap exists and is also an alternative to developing the payment function.

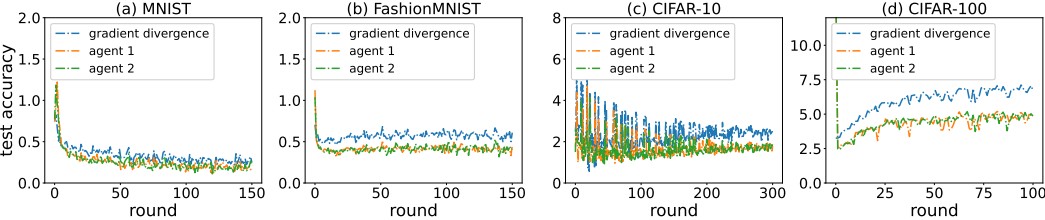

Figure 11: Gradient divergence with peers. The non-iid degree is 0.5.

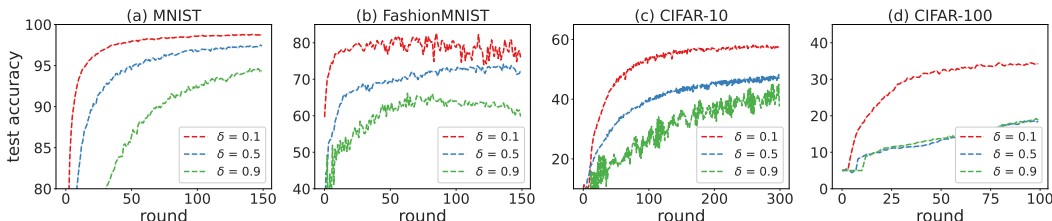

Figure 9: **(Long-tail distribution setting)** FL training process under different non-iid degrees.

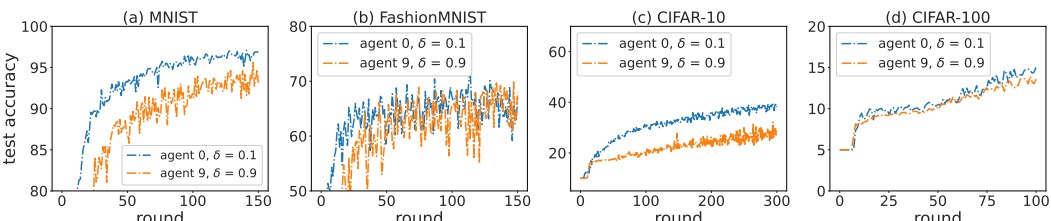

Figure 10: **(Long-tail distribution setting)** Performance comparison with peers. In default, the non-iid degree is 0.5.

