# OpenReview forum: "Incentivizing Data Collection from Heterogeneous Clients in Federated Learning"
_ICLR.cc/2024/Conference — Submitted to ICLR 2024_

### Official Review · Reviewer_H2b8 · 2023-10-27

**Soundness:** 2 fair
**Presentation:** 1 poor
**Contribution:** 2 fair
**Rating:** 3
**Confidence:** 4

**Summary:**

The paper studies how to incentivize the clients to contribute/share heterogeneous data in a federated learning setting. The authors consider the Wasserstein distance to formalize the data heterogeneity to derive a convergence bound, which is used to obtain a generalization gap between any two clients. Subsequently, the authors adopt the Stackelberg game to model the incentive process and derive a pure Nash equilibrium via the effort levels of the clients. Empirical results on the conventional FedAvg algorithm are provided.

**Strengths:**

- The paper studies a relevant and important problem in FL.

- The result for the pure Nash equilibrium (Theorem 5.2) provides a useful characterization of the properties of the utilities of the clients.

- The investigation of the constants (e.g., Lipshitz constant) in the results and assumptions is appreciated.

**Weaknesses:**

- The writing can be improved, for details see the questions.

- Some assumptions can be justified and motivated better.

- The main result (Lemma 4.2) is an application of an existing result, and is a looser upper bound, making its theoretical contribution not so clear.

- There seems to lack a comparison with existing methods or simple baselines.

**Questions:**

1. The abstract mentions "decentralized agents", but the setting does not seem decentralized since there is a central learner.

2. In introduction,
    > Hence the correct way to provide incentives for agents is to encourage agents to upload models that can capture the data heterogeneity.

    Is this made precise somehow?

3. In introduction,
    >  Such a solution is necessary for designing incentive mechanisms in a practical FL system but currently missing from all existing work ...

    Precisely, how is the necessity of such incentive mechanisms demonstrated? For instance, is it shown that all existing works perform sub-optimally in some way?

4. In introduction,
    > We are the first to prove that Wasserstein distance is an essential factor in the convergence bound of FL with heterogeneous data (Lemma 4.2)

    If my understanding is correct, the Wasserstein distance is used to derive a specific instantiation of an existing result, which already provides a way to formalize the heterogenity of data in FL. It is unclear why Wasserstein distance is essential.

5. Is the Wasserstein distance formally recalled somewhere? Is the definition of $\delta_k$ the formal definition? If so, does it mean that it does not work for regression problems?

6. In the definition of $\delta_k$, what is the support of $p^{(k)}, p^{(c)}$?

7. After the definition of $\delta_k$,
    > p(c) indicates the reference (iid) data distribution in the centralized setting

    Why is there a centralized setting, and what is meant by iid data distribution? Furthermore, how to obtain $\delta_k$ in implementation?

8. In Section 3.2,
    > Learner: seeking to train a classifier that endeavors to encourage agents to decrease the degree of non-iid. Thereby, attaining faster convergence at a reduced cost.

    What is meant by "decreasing the degree of non-iid"? Furthermore, precisely what is the cost in the "at a reduced cost"?

9. What are $ g_t, \bar{g}_t, f^* $ in Lemma 4.1?

10. In Lemma 4.2, what is the $L_{x|y=i}$-Lipschitz function Lipschitz w.r.t.~?


11. The learner is assumed to have access to $\mathcal{D}_c$, what are its properties and how can this assumption be satisfied?

12. What exactly is $\text{Bound}(e)$?

13. What is the "Ninety-ninety rule"?

14. In Theorem 5.1, the partial derivative of $f$ is taken. Does it mean you require the assumption that function $f$ is differentiable? If so, what are the (practical) motivations for this assumption?

15. In Definition 5.1, how are the constants $d_k^1, d_k^2$ set or determined, and how are $d_k^1, d_k^2$ related for a fixed $k$?


16. Why (only) Nash equilibrium in a Stackelberg game? Can something be said about the behavior of the learner (i.e., leader in the Stackelberg game) to consider the Stackelberg equilibrium?

17. In experiments under "Parameter analysis",
    > Recall that we set the learning rate $\eta = 0.01$.

    Is it for all experiments?

18. How should Figure 3 be interpreted? In particular,
    > the utilities of two randomly selected agents remain stable, which indicates an equilibrium.

    Which are the two agents? and the fluctuations (which the authors attribute to the random selection of the agents) make it difficult to see the equilibrium.

**Details Of Ethics Concerns:**

N.A.

---

> ### Author Response · Authors · 2023-11-16
> **Part 1: Rebuttal by Authors (Weaknesses and Question 1-4)**
>
> We want to thank the reviewer for their positive feedback and comments. We will address individual comments below.
>
>
>
> **Response to W1**: Please see our responses to ``Question 1-4, 7, 8, 18”.
>
> **Response to W2**: Please see our responses to ``Question 5, 11, 14”.
>
> **Response to W3**: We would like to clarify that we are not proposing a looser upper bound. Compared to Example 1's upper bound of the divergence term ( $4\eta_t^2(E-1)^2G^2$), our bound shown in Lemma 4.2 ($  64  \delta_k^2 (1+2\eta_t)^{2(E-1)} \eta_t^2 (E-1)  G^2 $) aligns consistently. We believe they are at least at the same tightness level.
> In this work, we mainly focus on exploring and understanding the impact of data heterogeneity on the convergence bound, instead of targeting a tighter upper bound or better proof techniques. The main of purpose of Lemma 4.2 is "we trying to build the connection between convergence and the Wasserstein distance”, where Wasserstein distance helps with quantifying the degree of non-iid in the local training data of an agent.
>
> **Response to W4**: Thank you for pointing out this. To our knowledge, there currently aren't any existing methods or baselines available that we can use for comparison in our study. We would be grateful for any recommendations you could provide for suitable baselines.
>
>
> **Response to Q1**: Sorry for any confusion. "Decentralized clients" in the context of federated learning or distributed systems refer to individual nodes, devices, or entities that operate independently of a central authority or location. Each client holds its own dataset and performs computations on this data locally. To prevent any misunderstandings in the revised version, we will refrain from using the term “decentralized”.
>
> **Response to Q2**: Thank you for pointing out this. We will revise it as “Hence, the appropriate method to incentivize agents should take into account not only the size of the sample, but also the potential data heterogeneity.”
>
> **Response to Q3**: A common limitation in existing studies is their focus on measuring each agent's contribution by the number of samples used, thereby encouraging the use of more data samples. These approaches tend to overlook the impact of data heterogeneity. Solely using sample size to evaluate contribution may not effectively reflect the quality of the data, especially in federated learning where data heterogeneity largely affects model convergence. Hence, an incentive mechanism that accounts for data heterogeneity is essential.
> From the learner's perspective, our proposed mechanism facilitates more optimal model convergence compared to existing methods. For example, at the same sample size level, our model's convergence is significantly faster compared to existing mechanisms. This improved performance is largely due to our approach effectively addressing data heterogeneity. This is evidenced by the performance results shown in Figure 1.
>
> **Response to Q4**: We admit that the notion of the Wasserstein distance has been applied in various fields, not just limited to federated learning. However, to our knowledge, there is no existing similar work that has specifically applied the Wasserstein distance to specifically assess the impact of data heterogeneity on the convergence bound in federated learning.
> In federated learning, "label shift" is the most common non-iid setting, which refers to a scenario where the distribution of labels (or output classes) in the training data varies across different clients over time. Therefore, the Wasserstein distance serves as a practical and important metric for measuring the label distribution, particularly in classification problems. This distance measures how much one distribution needs to be altered to resemble another, making it highly relevant for evaluating discrepancies in label distributions across different datasets. Apologies for any misunderstanding regarding the use of 'essential'. We will replace it with 'important'.

---

> ### Author Response · Authors · 2023-11-16
> **Part 2: Rebuttal by Authors (Question 5-9)**
>
> **Response to Q5**: Limited by paper space, we slightly introduce the Wasserstein distance in the definition of $\delta_k$.The Wasserstein distance is a well-known measure of the distance between two probability distributions. It is a concept derived from the field of optimal transport, where it represents the minimum "cost" required to transform one distribution into another. This metric provides a more intuitive way to compare distributions, consistent with our objective to reduce the degrees of non-iidness by adding additional samples.The 11-norm form of the Wasserstein distance is formally defined as $W_1(P, Q)= \inf _{\gamma \in \Gamma(P, Q)} \int\_{X \times Y} \|x-y\|_1 d \gamma(x, y)$, where $\|x-y\|_1$ is the l1-norm, calculating the absolute difference between points $x$ and $y$ in the distributions. We simply adapt this into a discrete distribution format, as illustrated in the definition of $\delta_k$.
> In our works, we mainly focus on classification problems, commonly encountered in federated learning. Therefore, the formulation of our $\delta_k()$ is defined over "I", which represents the number of classes in a classification problem.
>
>
>
> **Response to Q6**:  For classification problems,  the support of the discrete probability distributions $p^k$ and $p^c$ is the proportion for the entire set of possible classes within the classification problem.
>
> **Response to Q7**: This centralized setting refers to a scenario where data is aggregated in a single, central repository, in contrast to the distributed nature of federated learning where data is scattered across multiple clients. Typically, we would use the distribution of this ideal centralized setting $p^c$ as the iid distribution for federated learning and a benchmark to measure the degrees of non-iidness in clients' data, providing a reference point for comparing the data distribution across various decentralized clients.
> Owing to privacy concerns, accessing the iid distribution across agents is not possible. Thus, in federated learning, we often use the distribution $p^c$, derived from an ideal centralized setting, as the representative iid distribution. This serves as a standard for measuring the degrees of non-iidness in clients’ data, offering a basis for comparison across these distributed data sources. In practice, $p^c$ typically reflects the test distribution for performance evaluation from the learner's standpoint.
>
> To assess the degree of non-IIDness in an agent's local training data without it being explicitly provided, one potential approach is to use statistical techniques to analyze the data's distribution. Given the definition of $\delta_k = \frac{1}{2}\sum_{i=1}^{I}|p^{(k)}(y=i) - p^{(c)}(y=i)|$, in practice, we can reasonably assume that statistical information of local datasets (especially label proportions $p^{(k)}(y=i), \forall i \in [I]$), is accessible to the learner without compromising privacy. Also, the learner can disclose the label proportions of the reference dataset or distribution  $p^{(c)}(y=i), \forall i \in [I]$ to agents. This act effectively communicates a preference for iid settings to the agents. These assumptions allow for a straightforward assessment of the degrees of non-IIDness in the data.
>
> **Response to Q8**: "Decreasing the degree of non-iid" implies that agents are incentivized to modify their local datasets to resemble more iid-like distributions. Specifically, this can involve incentivizing agents to contribute more data from minority classes instead of majority ones in their datasets. "A reduced cost" refers to the learner's aim to minimize the expenses incurred while motivating agents to decrease the non-iid degree of their data, thereby maximizing the learner's payoff.
>
> **Response to Q9**: Apologies for missing explanations on these parameters. We will fix this in the revision.
> $f^* = f(w^*)$ indicates the optimal empirical risk.
> Then, $ \overline{\mathbf{g}}\_t = \sum\_{k=1}^N p\_k \nabla F\_k(\mathbf{w}_t^k) $ represents the weighted sum of gradients of loss functions $F_k$ for each client $k$. Here, $\mathbf{w}_t^k$ are the model parameters for client $k$ at time $t$, and $p_k$ is the weight for each client.
> Similar to $\overline{\mathbf{g}}\_t $, $\( \mathbf{g}\_t = \sum\_{k=1}^N p\_k \nabla F\_k(\mathbf{w}\_t^k, \xi\_t^k)$ includes a stochastic component $\xi\_t^k $, which typically represents a stochastic component such as a minibatch of data. In practice, $ \mathbb{E}\mathbf{g}_t = \overline{\mathbf{g}}_t $.

---

> ### Author Response · Authors · 2023-11-16
> **Part 3: Rebuttal by Authors (Question 10-18)**
>
> **Response to Q10**: There is no extra difference between $L\_{x|y=i}$-Lipschitz and typical $L$-Lipschitz. In our notation, we use subscripts solely for denoting the class. More specifically, $\nabla\_{\mathbf{w}} \mathbb{E}\_{x|y=i} [\ell(x,\mathbf{w})]$ is $L_{x|y=i}$-Lipschitz continuous,  i.e., $|| \nabla\_{\mathbf{w}} \mathbb{E}\_{x|y=i} [\ell(\mathbf{w}\_{t}^{k}, x)] -\nabla\_{\mathbf{w}} \mathbb{E}\_{x|y=i} [\ell(\mathbf{w}\_{t}^{k'}, x)] || \leq L\_{x|y=i} ||\mathbf{w}\_{t}^{k}-\mathbf{w}\_{t}^{k'}||$.
>
> **Response to Q11**: Actually, $D_c$ is an auxiliary validation set, following the test set's data distribution. It's common to assume access to such a validation set for evaluating model performance. In our study, this set is solely used for evaluation purposes, meaning no extra information is introduced by incorporating the validation set.
>
> **Response to Q12**: $\mathrm{Bound}(e) $ denotes the model's performance upper bound under agents' efforts $e$. For example, $\mathrm{Bound}(e) $ represents the upper bound as illustrated in Example 1.  More details can be found in the definition of payoff functions.
>
> **Response to Q13**: The ninety-ninety rule is a humorous aphorism that states: "The first 90 percent of the code accounts for the first 90 percent of the development time. The remaining 10 percent of the code accounts for the other 90 percent of the development time."
>
>
> **Response to Q14**: Yes, the payment function $f()$ is required to be differentiable. This characteristic is essential for the concept of well-behaved utility functions, as outlined in Definition 5.1. It's quite common to assume that utility functions are well-behaved, such as being convex or smooth, when analyzing Nash equilibrium[1, 2, 3].  With this property, we can demonstrate that the best response function has a fixed point, which then serves as the equilibrium in our framework.
>
> **Response to Q15**: Based on Assumption 5.1, we can determine the maximum and minimum values of $\partial f\left(e_k, e_{k^{\prime}}\right) / \partial e_k$ when $e_k=0$ and $e_k=1$, respectively. These extremal values correspond to the constants $d_k^1$ and $d_k^2$. More details can be found in the proof of Lemma B.1 of Appendix B.3.
>
> **Response to Q16**: In Remark 1 of Appendix B.3, we additionally explore the existence of other common equilibriums in our context, including no one participating and free-riding. Briefly speaking, the Stackelberg game in game theory is a model describing a strategic interaction where players move sequentially rather than simultaneously. It consists of a leader and one or more followers, where the leader must anticipate how followers will respond to maximize its payoff, while followers act to maximize their own payoffs given the leader's action. Hence, the structure of the Stackelberg game is indeed well-suited to address the incentive issues in federated learning.
>
> **Response to Q17**: Yes.
>
> **Response to Q18**: Sorry for this typo. From the theoretical aspect [Theorem 5.3], a Nash equilibrium in our settings means that no agent can improve their utility by unilaterally changing their invested effort levels. Hence, the utility of agents could be viewed as an indicator of the equilibrium $\rightarrow$ if the utility of agents remains almost constant as the training progresses, an equilibrium is reached. As shown in Figure 3, the utilities of two randomly selected agents remain stable, indicating equilibrium.
> We can attribute the observed fluctuations to randomness for two reasons. Firstly, Figure 4 demonstrates that agents can achieve maximum utility despite randomness, indicating the utility function's stability. Secondly, in Remark 2 of Appendix B.3, we also discuss the consistency between randomly selecting one peer agent and the extension that takes an average over all other agents. Ideally, this would result in the utility curves being flat, represented as horizontal lines.

---

> > ### Comment · Reviewer_H2b8 · 2023-11-21
> > **Response**
> >
> > I wish to thank the authors and praise their efforts for the detailed response.
> >
> > [**Regarding W2.**]
> > The new technical result does seem to be on the same order as the existing one, though the authors also mention that it is *not* shown to be tighter and believed to be as tight. Making this very clear would be helpful, especially since the inequality used in the proof only tells the reader that the existing result is no looser.
> >
> > [**Regarding W4.**]
> > For instance, (Lin et al., 2023) seems a relevant baseline. While it may be that this baseline is very recent and the authors missed it, I still believe that some simple and intuitive baselines are possible: (i) if an agent never changes the contribution regardless of the value; (ii) an agent that monotonically increases the contribution based on the value (and this monotonicity can be varied in the experiments). These are by no means exhaustive and I am not asking for more results during the short remaining period of the rebuttal because it does indeed take time to obtain such results.
> >
> > As a remark, it is expected that no relevant work would fit perfectly in the setting considered here, which is why the reader might expect to see that the proposed method out-performs such baselines in the metric that this paper is focusing on.
> >
> > [**Regarding Q4.**]
> > It does seem that (Huang et al., 2023) use Wasserstein distance for contribution evaluation of the clients, supported with a proven convergence bound. (Farnia et al., 2022) also appear to be utilizing the optimal transport in a similar way.
> >
> > [**Regarding Q7.**]
> > The proposed method requires a key assumption on the access to the information that is the label proportion of each local dataset. This can be difficult realize in practice and thus become limitations of the method: (i) how to show that this information does not compromise privacy; (ii) how to show that the clients are willing to provide such information truthfully.
> >
> > [**Regarding Q11.**]
> > That the learner (or server) has access to some validation dataset is an assumption, which is difficult to guarantee in practice (Soltani et al., 2023). It is indeed a technical challenge several works try to explicitly address.
> >
> > [**Regarding Q13.**]
> > The meaning of the phrase was not particularly clear or and its usage not particularly formal.
> >
> >
> > [**Regarding Q14.**]
> > This assumption should be made explicit, as with other previously described assumptions.
> >
> >
> > Some of my questions are answered, and the authors took note of some the typos and incorrectly used phrases. I thank the authors again for their efforts in the response. However, there remain some concerns, including comparison with baselines and assumptions (that are not clearly highlighted in the paper but quite important), and my rating remains unchanged.
> >
> >
> > **References**
> >
> > Xiaoqiang Lin, Xinyi Xu, See-Kiong Ng, Chuan-Sheng Foo, and Bryan Kian Hsiang Low. 2023. Fair yet asymptotically equal collaborative learning. In ICML.
> >
> > Huang, J., Hong, C., Liu, Y., Chen, L.Y., Roos, S. (2023). Maverick Matters: Client Contribution and Selection in Federated Learning. In: Kashima, H., Ide, T., Peng, WC. (eds) Advances in Knowledge Discovery and Data Mining. PAKDD 2023. Lecture Notes in Computer Science, vol 13936. Springer, Cham.
> >
> > F. Farnia, A. Reisizadeh, R. Pedarsani and A. Jadbabaie, "An Optimal Transport Approach to Personalized Federated Learning", in IEEE Journal on Selected Areas in Information Theory, vol. 3, no. 2, pp. 162-171, June 2022.
> >
> > Behnaz Soltani, Yipeng Zhou, Venus Haghighi, John C. S. Lui. A Survey of Federated Evaluation in Federated Learning. 2023. In IJCAI.

---

### Official Review · Reviewer_6Cg7 · 2023-10-31

**Soundness:** 2 fair
**Presentation:** 2 fair
**Contribution:** 2 fair
**Rating:** 5
**Confidence:** 2

**Summary:**

The paper proposes an FL framework that incentivizes clients to collect high-quality data. The framework is based on the principle of mechanism design, where the server designs a payoff function that takes as input the clients' "effort" level and returns a payoff to each individual client. The paper assumes that the cost of a client is some value proportional to the iid-ness of their local dataset (i.e. how similar it is to other clients' datasets). Under this condition, they propose using a payment function based on a client's performance relative to its peers. They derive theoretical results and and run experiments showing the utility of this payment function for various clients.

**Strengths:**

The paper is a novel way to view FL as a mechanism design problem between data and model providers.
The paper emphasizes the importance of having clients collect non-iid data for FL and highlights the cost of collecting this data.

**Weaknesses:**

The experimental results are very limited. The first two plots simply show that training on a limited set of labels is harmful. The other two plots show that utility is stable across rounds.

Furthermore, it is not clear why stable utility across rounds indicates the strategies are at an equilibrium? In my view the variation across rounds does not seem relevant. The important part is in Figure 4 which shows the implicit cost-reward tradeoff from varying the client's iid-ness.

The FL setup itself is also limited in that it resamples data from the same shared pool. While this toy setup might provide the clearest results it is also quite unrealistic. If FL clients are the ones who are generating the data, then clients will usually have distinct examples.

The time-varying aspects of the paper are not clear. Do the clients adjust their dataset over the course of training?

**Questions:**

If we can account for the noise in client sampling to compute the payment (e.g. take an average over all other clients), would the utility curves (Fig. 3,4) be horizontal lines?

It seems surprising that utility is stable across rounds, despite the fact that training is often noisy. Wouldn't this noise also be reflected in the relative performance between two client models?

---

> ### Author Response · Authors · 2023-11-16
> **Rebuttal by Authors**
>
> We want to thank the reviewer for their positive feedback and comments. We will address individual comments below.
>
>
> **Response to W1**: We would like to clarify that the four plots presented in Section 6 comprehensively demonstrate our work's core contributions. These plots not only validate the theoretical analysis detailed in Section 4 but also effectively evaluate the proposed incentive mechanism.
>
> Our theoretical framework forms the basis of our study, especially in validating Theorem 4.3, which is crucial for developing score functions to incentivize agents. Figures 1 and 2 empirically validate the heterogeneity in performance and the disparity among agents, consistent with our theoretical insights.
>
> Additionally, Theorem 5.3 indicates that a Nash equilibrium leads to stable utilities against individual effort changes, as shown by the steady utilities in Figure 3, indicating equilibrium. Figure 4 further illustrates the impact of effort changes on utility, emphasizing our incentive mechanism's effectiveness. This also demonstrates the greater stability of the utility function. Despite the variation caused by random peer selection, the utility at optimal effort levels is higher on average compared to other levels.
>
> In Appendix C, we also provide additional experimental results. Specifically, we demonstrate heterogeneous efforts using another common metric for non-iid data: the number of classes in local datasets. Additionally, we conduct evaluations using a standard data partitioning approach, where the entire dataset is evenly divided among agents, followed by sampling from these local datasets. These corresponding performance results are consistent with the previous results.
>
> **Response to W2**: Given Theorem 5.3, a Nash equilibrium in our settings means that no agent can improve their utility by unilaterally changing their invested effort levels. Consequently, the utility of agents can serve as an important measure to indicate or confirm the presence of an equilibrium. As shown in Figure 3, the utilities of two randomly selected agents remain stable, which indicates an equilibrium. Specifically, the existing variation of utility mainly results from the randomness of selecting peers.
>
> **Response to W3**: Thank you for highlighting this. It's possible for agents to possess similar data samples, especially when sourcing data from the internet. For example, this similarity can often occur in situations where agents collect publicly available data, such as commonly used datasets or information from widely accessed web sources. In Appendix C.2, we also extend our performance evaluation using other non-iid metrics, like the number of classes. Additionally, we consider a general setup where the entire dataset is evenly partitioned among agents, followed by sampling from these local datasets, to further validate our findings.
>
> In response to your concern, we are conducting more experiments following the same data partitioning setup as outlined in Appendix C.2. However, we are considering a modification: instead of the uniform distribution for minority classes as used in Appendix C.2, we're looking at implementing long-tailed or exponential distributions for these classes. This change is aimed at amplifying the non-iid characteristics in our experiment. These results will be updated to the revision before the rebuttal deadline.
>
>
> **Response to W4**: Yes, clients will resample data from the entire dataset at each round to compose their local datasets, while maintaining the same level of non-IIDness.
>
>
> **Response to Q1**: Note that the introduced randomness (randomly selected peers) serves as an effective tool to circumvent the coordinated strategic behaviors of most agents. In Remark 2 of Appendix B.3, we also discuss the consistency between randomly selecting one peer agent and the extension that takes an average over all other agents. Ideally, in such a scenario, the utility curves for the agents would appear as horizontal lines.
>
> **Response to Q2**: Utility is not dependent on training performance, as it is derived from theoretical calculations. For clarity, in this paper, the utility function is defined as $u_k(e_k) \triangleq \mathrm{Payment}_k(e_k) - \mathrm{Cost}_k(e_k) =\frac{1}{\mathrm{ Upper\_Bound}(F_c(\mathbf{w}^k)- F_c(\mathbf{w}^{k'})) } - c_k \cdot d(|\delta_k(0)- \delta_k(e_k)|)$. It represents the collective effort level of agents, which correlates more closely with their final performance. Therefore, the noise indeed affects the relative performance between two client models but would not influence the computed utility.

---

> ### Author Response · Authors · 2023-11-19
> **Additional experiments for weakness 3**
>
> Thank you for your patience. We have completed the additional experiments as previously discussed, using long-tailed distributions for minority classes as outlined in Appendix C.2. The corresponding results shown in Figures 7–10 are consistent with those illustrated in Figures 1–2. The results have been updated in the revised manuscript, addressing your concerns. Thank you for your patience and valuable feedback.

---

> > ### Author Response · Authors · 2023-11-23
> >
> > We thank the reviewer for those useful comments, and we have incorporated all the discussion and experiment results from the rebuttal into our paper. We hope the reviewer finds our response and revision to the manuscript satisfactory. If the reviewer has any additional suggestions or comments, we are more than happy to address them and further revise our manuscript!

---

### Official Review · Reviewer_2yXg · 2023-11-01

**Soundness:** 3 good
**Presentation:** 3 good
**Contribution:** 3 good
**Rating:** 6
**Confidence:** 2

**Summary:**

The paper proposes an incentive-aware framework to encourage the participation of clients with a view to using data heterogeneity measures to accelerate the convergence of the global model.
A key component of this framework is the use of wasserstein distance to quantify the degree of non-IIDness in the local training data of a client. In addition, the framework includes the design of reward functions that use the generalization error gap between two agents to induce more truthful reporting.
The paper combines all these ideas and presents it as a two-stage Stackelberg game, proving the existence of an equilibrium within this setting.

**Strengths:**

- The presentation of the concept, with detailed explanations of base terminology and notations, examples and relation to broader goals discussed throughout the paper is well thought out.
- A clear and unambiguous statement of the overall goal of the paper and effort detailing how it differs from existing approaches offers a clear picture with regard to the current status of the domain.

**Weaknesses:**

- While there are a number of key assumptions made throughout the paper, an important one is "equilibrium shall prevail when the impact of unilateral deviations in an agent's effort level remain limited to their own utility." and an equilibrium solution may not exist if small changes affect other agent's utilities. Could the authors discuss whether adversarial behavior falls under this category of assumptions and how can the system potentially protect itself from such cases.
- A recent trend in federated learning involves the use of pre-trained weights, either from ImageNet pretraining or foundation models, to close the gap in performance between IID and non-IID scenarios. Given that such choices directly affect the prediction ability of agents, with each agent having a unique understanding of the dataset based on the training setup and choice of pre-trained model, could the authors comment on pre-trained weights suppressing the underlying heterogeneity of local data distributions and how this affects the proposed framework?
- Could the authors discuss how best to assess the degree of non-IIDness of an agent's local training data ($\delta_k$) without it being provided?

**Questions:**

For key questions, please refer to the weaknesses section.

EDIT: I appreciate the detailed responses provided by the authors. Taking into the consideration the new insights provided by the authors' feedback, I believe that the current manuscript serves as a good starting point in studying incentive mechanisms in FL. In addition, the current submission also highlights the need to bridge theoretical insights with the current SOTA FL methodologies.
Considering all these points together, I plan to maintain my original score.

---

> ### Author Response · Authors · 2023-11-16
> **Rebuttal by Authors**
>
> We want to thank the reviewer for their positive feedback and comments. We will address individual comments below.
>
> **Response to W1**: Indeed, the above-mentioned statements are mild assumptions that can be reformulated as properties of utility functions. It's quite common to assume that utility functions are well-behaved, such as being convex or smooth, when analyzing Nash equilibrium [1, 2, 3]. For instance, in Definition 5.1, we formulate them as the bounds on the first-order gradient of the utility function, which are conditions that are generally easy to meet. Given this framework, discussing adversarial behaviors becomes less pertinent.
> Indeed, the Nash equilibrium acts as a safeguard against adversarial behaviors. It ensures that agents can maximize their utilities exclusively by exerting the optimal level of effort. This aspect highlights a significant advantage of the Nash equilibrium concept.
>
> **Response to W2**: Thanks for this great question! Utilizing pre-trained weights in federated learning effectively suppresses the impact of data heterogeneity, resembling warm-start strategies or personalization. However, the influence of this heterogeneity could be reserved during continued training from these weights. It's important to highlight that theoretical analysis serves as the foundational starting point for our work. The theoretical framework might differ in the context of pre-trained weights. To integrate this scenario into our proposed model, we might need to introduce certain assumptions regarding the pre-trained weights.
>
>
> **Response to W3**: [Pre-effort] To assess the degree of non-IIDness in an agent's local training data without it being explicitly provided, one potential approach is to use statistical techniques to analyze the data's distribution. Given the definition of $\delta_k = \frac{1}{2}\sum_{i=1}^{I}|p^{(k)}(y=i) - p^{(c)}(y=i)|$, in practice, we can reasonably assume that statistical information of local datasets (especially label proportions $p^{(k)}(y=i), \forall i \in [I]$), is accessible to the learner without compromising privacy. Also, the learner can disclose the label proportions of the reference dataset or distribution  $p^{(c)}(y=i), \forall i \in [I]$ to agents. This act effectively communicates a preference for iid settings to the agents.These assumptions allow for a straightforward assessment of the degrees of non-IIDness in the data.
>
> [Post-effort] Besides, from the perspective of the learner, in the context of a complete-information game, the learner can assess post-effort $\delta_k$ via the Nash equilibrium.
>
>
> [1] One for one, or all for all: equilibria and optimality of collaboration in federated learning, ICML 2021.
>
> [2] Mechanisms that incentivize data sharing in federated learning, arXiv preprint, 2022.
>
> [3] Collaboration equilibrium in federated learning, SIGKDD 2022.

---

### Official Review · Reviewer_WQhg · 2023-11-05

**Soundness:** 3 good
**Presentation:** 2 fair
**Contribution:** 2 fair
**Rating:** 5
**Confidence:** 3

**Summary:**

The authors focus on decreasing data heterogeneity and training a better global model from the perspective of incentivizing data collection. The authors also propose a framework of the two-stage Stackelberg game to incentivize the data collection.

**Strengths:**

1.	This paper extends the convergence bound with the Wasserstein distance.
2.	The experiment results are convincing.

**Weaknesses:**

1. The model is not clear/justified what does the "effor" mean for agents? How can agents narrow down the non-iid degree of their own data simply by incurring more "effort"? I don't think such model of effort and non-iid level is reasonable and realistic.

2. The authors assume complete information, e.g., agents know the true central distribution, this is also not reasonable/justified.

3. Definition 5.1 is quite similar to Definition 6 in Blum et.al, 2021, but it is not cited when Definition 5.1 is proposed.

4. Not clear why Figure 3 indicates equilibrium.

**Questions:**

see weakness.

---

> ### Author Response · Authors · 2023-11-16
> **Rebuttal by Authors**
>
> **Response to W1**: In our paper, the term "effort $e_k$" refers to the level of effort exerted by a client in collecting additional data samples, with the primary objective of maximizing utility.
> In fact, agents can decrease the non-iid degree of their data by contributing their data with a preference for an IID setting. For example, agents can be incentivized to provide data samples for minority classes rather than for majority classes in their local datasets.
> It's common and rational to incentivize agents to offer more/high-quality data, known as data sharing. From the perspective of data-sharing, agents may exert effort to supply more data samples [1, 2]. Our work advances the field by establishing a preferred method for data collection in data-sharing.  Therefore, from other perspectives, agents are incentivized to use their data strategically rather than merely contributing all their local data immediately.
>
>  **Response to W2**: Though it seems less practical than the incomplete game, studying complete information games is a standard and key practice that provides vital theoretical insights into strategic interactions. Besides, the complete game is reasonable because most of the complete information is available [1, 2]. In fact, agents only know the class proportion of the true central distribution, rather than full information about the true central distribution. Therefore, we believe that accessing the statistics information of the true central distribution is a mild and reasonable assumption. For example, participants generally know the upper bounds of convergence and generalization loss, with only the marginal cost remaining private. This setup also makes sense when costs, like those for data creation (e.g., autonomous driving) or labeling, can be readily estimated by all parties.
>
>   **Response to W3**:  Thank you for pointing out this. We will fix this in the revision.
>
>   **Response to W4**: From the theoretical aspect [Theorem 5.3], a Nash equilibrium in our settings means that no agent can improve their utility by unilaterally changing their invested effort levels. Hence, the utility of agents could be viewed as an indicator of the equilibrium $\rightarrow$ if the utility of agents remains almost constant as the training progresses, an equilibrium is reached. As shown in Figure 3, the utilities of two randomly selected agents remain stable, which indicates an equilibrium. Note that the existing fluctuation of utility mainly results from the randomness of selecting peers.
>
>
> [1] Incentive Mechanisms for Federated Learning: From Economic and Game Theoretic Perspective, IEEE TCCN, 2022.
>
> [2] An Incentive Mechanism for Cross-Silo Federated Learning: A Public Goods Perspective, IEEE INFORCOM 2022.

---

> > ### Author Response · Authors · 2023-11-23
> >
> > We thank the reviewer for the useful suggestion, and we have incorporated all the discussion and experiment results from the rebuttal into our paper. We hope the reviewer finds our response and revision to the manuscript satisfactory. If the reviewer has any additional suggestions or comments, we are more than happy to address them and further revise our manuscript!

---

### Meta-Review · Area_Chair_ovrN · 2023-12-05

**Metareview:**

This paper proposes an incentive-aware framework for agent participation that considers data heterogeneity to accelerate convergence and presents a two-stage Stackelberg game to model the process.


**STRENGTHS**

(1) The problem is important.

(2) This paper has contributed some useful results, both theoretically and empirically, for this problem.


**WEAKNESSES**

(1) A major concern of this paper is a serious lack of clarity of presentation, such as identifying the limitations/assumptions of this work, several undefined notations, among others. As an example, several reviewers were unclear about how to assess the degree of non-IIDness of an agent's local training data and why it is possible to reduce it in practice. The authors have provided brief clarifications in their rebuttal, which we would like to see being elaborated further in their revised paper.

(2) There is also a lack of empirical comparison with existing works and simple baselines.

Hence, this paper requires a major revision.

We strongly encourage the authors to revise their paper according to the feedback above and in the reviewers' comments.

**Justification For Why Not Higher Score:**

There is a serious lack of clarity for this paper, which requires a major revision. This paper also lacks empirical comparison with existing works and simple baselines. See meta-review for more details.

**Justification For Why Not Lower Score:**

N/A.

---

### Decision · Program_Chairs · 2024-01-16

Reject